# Redirection for Erasing Memory (REM): Towards a universal unlearning method for corrupted data

**Stefan Schoepf**[*]
University of Cambridge

**Michael C. Mozer**
Google DeepMind

**Nicole Mitchell**
Google Research

**Alexandra Brintrup**
University of Cambridge

**George Kaissis**
Google DeepMind

**Peter Kairouz**
Google Research

**Eleni Triantafillou**
Google DeepMind

## Abstract

Machine unlearning is studied for a multitude of tasks, but specialization of unlearning methods to particular tasks has made their systematic comparison challenging. To address this issue, we propose a conceptual space to characterize diverse corrupted data unlearning tasks in vision classifiers. This space is described by two dimensions, the *discovery rate* (the fraction of the corrupted data that are known at unlearning time) and the *statistical regularity* of the corrupted data (from random exemplars to shared concepts). Methods proposed previously have been targeted at portions of this space and, as we show, fail predictably outside these regions. We propose *Redirection for Erasing Memory (REM)*, whose key feature is that corrupted data are redirected to dedicated neurons introduced at unlearning time and then discarded or deactivated to suppress the influence of corrupted data. REM performs strongly across the space of tasks, in contrast to prior SOTA methods that fail outside the regions for which they were designed.

## 1 Introduction

Unlearning is the problem of removing the effect of a subset of training data from a trained model (Nguyen et al., 2022). In this work, we consider a scenario where after having already trained a model on a dataset, we discover that a subset of the training data was accidentally mislabelled, low quality, or manipulated by an attacker, causing the model to make mistakes or exhibit unwanted behaviours. The goal of unlearning (UL) is to post-process the trained model to efficiently remove (the effect of) that *corrupted data* in order to restore the correct predictions and behaviours.

While the problem of unlearning has attracted significant attention (Triantafillou et al., 2024; Hayes et al., 2024; Goel et al., 2024), and a plethora of methods have been proposed, there is still a lack of scientific understanding of the behaviours of these methods on different types of UL tasks, with only early work in this direction (Zhao et al., 2024; Goel et al., 2024). This lack of an understanding when established unlearning methods fail or succeed is a fundamental blind spot as it hinders research progress and can lead to unpredictable failure in practice, as shown in this work.

To address this, our first contribution is a taxonomy of tasks for unlearning corrupted data shown in Fig. 1. Our taxonomy is based on the identification of two dimensions along which the behaviours of unlearning algorithms differ substantially. The first is the *discovery rate*, the fraction of corrupted data that have been identified and can be utilized by the unlearning algorithm to remove the effect of corruptions. Goel et al. (2024) previously studied the effect of discovery rates on unlearning performance, finding that performance of algorithms developed for the full discovery case drops off suddenly as the discovery rate is lowered continuously. We take this study a step further by identifying a second dimension that affects performance significantly, both on its own and through interaction with the discovery rate. The second dimension is the *statistical regularity* of the cor-

---
[*]Work done during the author's internship at Google DeepMind. PhD funded by EPSRC DTP [EP/W524633/1]. Corresponding author: ss2823@cam.ac.uk

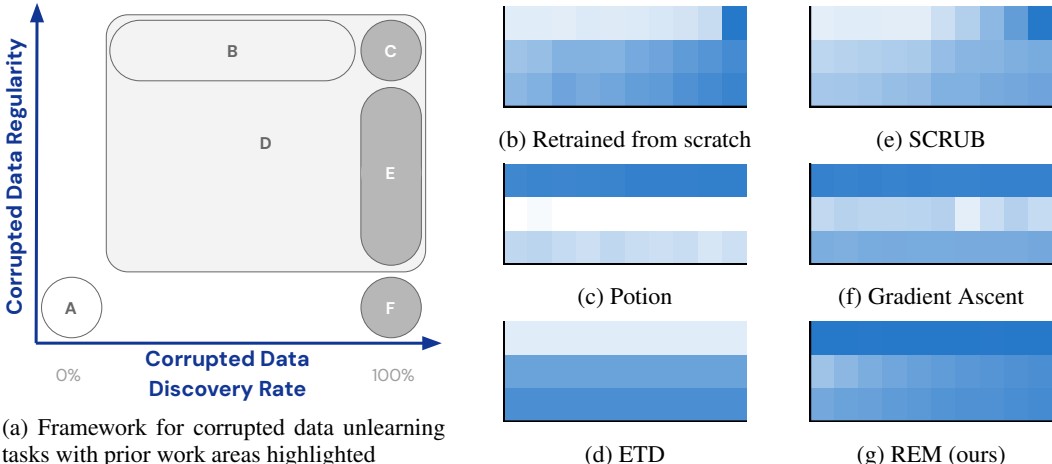

Figure 1: We present a new taxonomy of unlearning tasks in terms of two dimensions: the regularity and the discovery rate of the corrupted data we wish to unlearn. The highlighted areas in (a) show tasks studied in prior work. Subplots (b-e) show an aggregate metric of unlearning performance (see Section 2) of methods for different discovery rates >0% (x-axis) and regularities (y-axis), instantiated via the benchmarks of (Goel et al., 2024). Darker color is better. Prior methods succeed only in slices of this 2D space, mainly failing along the regularity axis. Representative prior work: [A] Maini et al. (2023), [B] Schoepf et al. (2024b), [C] Chundawat et al. (2023); Kurmanji et al. (2023), [D] Goel et al. (2024), [E] Zhao et al. (2024); Foster et al. (2024); Chundawat et al. (2023); Kurmanji et al. (2023), [F] Chundawat et al. (2023); Kurmanji et al. (2023)

rupted data, tracing a spectrum from "spurious" corruptions (such as random mislabeling), towards structured corruptions that systematically affect related data points (such as a poison trigger that redirects all images on which it appears to a pre-specified class label). Our key finding is that, in our 2D space of tasks, different state-of-the-art (SOTA) methods succeed in different slices, but each fails predictably and catastrophically everywhere else. These methods are therefore risky to use in practice when the task specification is not known.

To address the need for a universal method for unlearning corrupted data that covers the 2D space, we present our second contribution: *Redirection for Erasing Memory (REM)*, a new unlearning algorithm that is the first to perform strongly across our task space. REM employs a novel mechanism, inspired by the memorization mitigation method *Example-Tied Dropout (ETD)* (Maini et al., 2023), to redirect the corruption to a dedicated part of the network (newly initialized capacity added to the model by REM) that can then be conveniently dropped off or deactivated to unlearn even undiscovered corrupted data of any regularity. We benchmark REM against SOTA unlearning methods as well as ETD on low, medium, and high regularity unlearning tasks for discovery rates ranging from 10% to 100%. Experimental results on CIFAR10 (Krizhevsky et al., 2009) and Street View House Numbers (SVHN) (Netzer et al., 2011) with ResNet-9 (He et al., 2016) and Vision Transformer (ViT) (Dosovitskiy et al., 2021) as well as different optimizers show that REM is the only method to perform well or equally well across the entire 2D space.

## 2 PROBLEM DEFINITION AND BACKGROUND

Machine unlearning applications range from privacy protection of individuals (Neel et al., 2021; Sekhari et al., 2021; Hayes et al., 2024) to correcting errors in the model due to corrupted training data (Goel et al., 2022; Kurmanji et al., 2023; Goel et al., 2024), each of which has different goals and evaluation metrics. In this work, we focus on unlearning corrupted data for classification tasks. Practical examples of corrupted data can range from simple data entry errors (Tanno et al., 2022; Schoepf et al., 2024a) to large scale data poisoning attacks (Carlini et al., 2024; Zhang et al., 2024b).

**Problem definition.** Let $\mathcal{D}_{tr}$ denote a training dataset, and $\mathcal{D}_{test}$ a held-out dataset of the same distribution. Further, let $\mathcal{A}$ denote a training algorithm, and $\theta = \mathcal{A}(\mathcal{D})$ the weights of a neural

network that are learned by applying $\mathcal{A}$ on $\mathcal{D}$. We consider scenarios where some training data $\mathcal{D}_c \subset \mathcal{D}_{tr}$ is corrupted. Let $\mathcal{D}_f \subseteq \mathcal{D}_c$ denote the *forget set* comprised of corrupted data discovered after training. We refer to the rest of the training data, $\mathcal{D}_{tr} \setminus \mathcal{D}_f = \mathcal{D}_r$ as the *retain set*. This formulation reduces to the most commonly-studied variation of unlearning when $\mathcal{D}_f = \mathcal{D}_c$; i.e. the full set of data that we wish to unlearn is provided in the forget set. In that case, a straightforward but computationally expensive solution is to retrain from scratch to obtain a new model $\theta_r = \mathcal{A}(\mathcal{D}_r)$ that is not affected by corrupted data. For "partial discovery", defined as $\mathcal{D}_f \subset \mathcal{D}_c$, $\theta_r$ is not a solution since undiscovered corrupted data $\mathcal{D}_c \setminus \mathcal{D}_f = \mathcal{D}_{undiscovered}$ is part of $\mathcal{D}_r$, so re-training on it does not eliminate the corruptions (Goel et al., 2024). The problem we are interested in is to design a ($\mathcal{A}$, $\mathcal{U}$) pair of a learning algorithm $\mathcal{A}$ and post-processing (unlearning) algorithm $\mathcal{U}$ such that, when a forget set $\mathcal{D}_f$ is identified, $\mathcal{U}$ can *efficiently* post-process the trained model $\theta_o = \mathcal{A}(\mathcal{D}_{tr})$ to yield an unlearned model $\theta_u = \mathcal{U}(\theta_o, \mathcal{D}_f)$ that does not suffer from failure modes caused by $\mathcal{D}_c$, while having high utility. Note that some choices of $\mathcal{A}$ may facilitate the success of $\mathcal{U}$ to post-process for unlearning. We assume for simplicity that $\mathcal{U}$ can access $\mathcal{D}_{tr}$, as is common in the literature.

**Memorization and statistical regularities.** Feldman (2020) refers to an example $x$ as being "memorized" by a learning algorithm $\mathcal{A}$ in a training set $\mathcal{D}_{tr}$ if, in expectation, models obtained through recipe $\mathcal{A}(\mathcal{D}_{tr})$ are much more likely to make correct predictions on $x$ compared to models obtained through recipe $\mathcal{A}(\mathcal{D}_{tr} \setminus x)$. Intuitively, $x$ is memorized if it is required in $\mathcal{D}_{tr}$ to be correctly predicted. This will be the case for atypical / irregular examples (Feldman & Zhang, 2020), whereas typical / regular examples that are similar to many others in the dataset may be predicted correctly simply due to generalization (i.e. being learned through other samples), without having to be part of the training set. Similarly, Jiang et al. (2020) define a *consistency score* of an example $x$ to measure its statistical regularity via the expected accuracy on that example under models obtained by training on different subsets of $\mathcal{D}_{tr}$ that exclude $x$. Building upon this notion, we identify the statistical regularity of the corrupted data as a key factor influencing behaviours of unlearning algorithms.

**Example-Tied Dropout (ETD)** is a training recipe proposed by Maini et al. (2023) that separates model parameters into those that intend to capture "generalizable information" that is shared between many data points, and those that intend to "memorize" information that is specific to one or a few data points. This separation is made within every layer of a neural network. During training, the computational paths of all data points pass through all of the generalization neurons (that is, the *generalization neurons* are never dropped out), but when it comes to the memorization partition, each example only passes through its dedicated path, based on an example-tied dropout mask that is randomly determined ahead of training and then fixed. This allows different pathways in the memorization partition to be "owned" by different examples, enabling those examples to encode their potential irregularities in their dedicated paths. On the other hand, example-specific irregularities won't be as easily encoded in the *generalization neurons*, since those are shared by all examples. At inference time, the memorization neurons are dropped out, reducing or eliminating example-specific information. A similar approach has been applied by Ghosal et al. (2025) to prevent verbatim memorization in LLMs. We propose ETD as an UL algorithm, which can be seen as setting $\mathcal{A}$ as ETD training followed by dropping out all memorization neurons post training, and setting $\mathcal{U}$ a no-op.

## 3  RELATED WORK

**Unlearning methods.** We build upon the benchmarking selection of Goel et al. (2024) to cover different state-of-the art approaches to unlearning and extend it with recent advances and overlooked baselines. Goel et al. (2024) included Bad Teacher (BadT) (Chundawat et al., 2023), SCRUB (Kurmanji et al., 2023), Selective Synaptic (SSD) (Foster et al., 2024), fine-tuning, and retraining from scratch. BadT randomizes labels on $\mathcal{D}_f$ by distilling from a randomly initialized network to induce forgetting and distilling the remainder of the samples $\mathcal{D}_{tr} \setminus \mathcal{D}_f$. SCRUB alternates between a step of distillation away from the original model on $\mathcal{D}_f$ and a step towards the original model on $\mathcal{D}_r$ for model utility preservation on the remaining data. SSD determines disproportionately important parameters for $\mathcal{D}_f$ compared to $\mathcal{D}_{tr}$ in the model via the Fisher Information Matrix and dampens these parameters to induce unlearning. We replace SSD in our benchmarks with Potion (Schoepf et al., 2024b) which builds upon SSD and is the SOTA in poison unlearning as defined by Goel et al. (2024) (see Fig. 1). Potion iteratively increases the number of parameters in the model that it modifies analogously to SSD until unlearning of the poison trigger occurs. Potion stops unlearning as soon as it detects that the accuracy of the model on the forget set drops below a preset threshold,

indicating that the poison has been unlearned. Fine-tuning continues to train the model on only $\mathcal{D}_r$, relying on catastrophic forgetting to remove $\mathcal{D}_f$. Retraining from scratch trains a new model using only $\mathcal{D}_{tr} \setminus \mathcal{D}_f$. Ascent performs gradient ascent on $\mathcal{D}_f$. NPO (Zhang et al., 2024a) is an alignment-inspired loss function for unlearning that overcomes the catastrophic collapses of Gradient Ascent. This is achieved via a reference-model based loss calculation.

**Investigation of unlearning limitations.** Goel et al. (2024) studies the effect of partial discovery on unlearning method effectiveness as shown in Fig. 1. For unlearning poisoned data, they demonstrate that all methods apart from Foster et al. (2024) fail to remove the effect of the poison when only a subset of the poisoned examples is given (partial discovery). They also consider another setup, inter-class confusion, where samples from two classes are swapped with each other, where no method shows satisfactory performance in the partial discovery case. Zhao et al. (2024) identify interpretable characteristics of forget sets that substantially affect the difficulty of the problem, but they don't consider the statistical regularity of the forget set in their investigation. Rezaei et al. (2025) show that standard unlearning methods designed for erasing knowledge may fail at *restoring* knowledge present prior to corruption. This relates to our metrics of healing, where we are interested in redirection to the correct label, rather than just avoiding predicting the incorrect (e.g. poisoned) label. However, their investigation is in the context of LLMs and they don't establish a distinction between tasks of different discovery rates or regularities.

## 4 TAXONOMY OF UNLEARNING TASKS AND METHOD LIMITATIONS

**Taxonomy.** Our first contribution is to propose a new taxonomy of unlearning tasks according to two dimensions: the *discovery rate* and the *statistical regularity* of the corrupted data that we wish to unlearn. The discovery-rate dimension was previously discussed and empirically investigated in Goel et al. (2024) but is insufficient to explain model failures, as methods such as Potion (Schoepf et al., 2024b) excel at some low discovery tasks but then fail at other full-discovery tasks. We further identify regularity as a key dimension that affects the behaviour of unlearning algorithms both on its own and in conjunction with the discovery rate. Our key observation is that, as shown in Fig. 1, each prior method may excel on a slice of our 2D space of unlearning tasks but fails catastrophically everywhere else. We discuss the limitations of different algorithms below.

**Discovery rate.** Unlearning methods that are designed for the traditional unlearning setting of full discovery fail catastrophically for lower discovery percentages as described by Goel et al. (2024). Retraining from scratch and unlearning methods that utilize the retain set, e.g. SCRUB, fail due to requiring a clean retain set for utility preservation, but in the partial discovery case, the retain set is contaminated and thus leads to reintroduction of corruptions (see Fig. 1 (b, e)). Methods such as Potion and SSD that only use $\mathcal{D}_f$ cause significant model utility damage due to the missing repair step on $\mathcal{D}_r$ (Goel et al., 2024; Foster et al., 2024).

**Regularity** refers to the self-similarity of the corrupted data $\mathcal{D}_c$. Informally, we say that a corruption has low regularity if the corrupted data does not share any structure or common patterns. For example, a random mislabelling of a randomly-chosen subset of examples is a low-regularity corruption because the self-similarity between samples that end up with the same label change is low. On the other hand, a poisoning attack that introduces a trigger to each image whose label it wants to redirect results in a high-regularity corruption since all corrupted images share the visual pattern of the poison trigger and are all labeled in the same way. In the context of the commonly studied problem of full discovery unlearning, class unlearning (Golatkar et al., 2020) and random subset unlearning (Chundawat et al., 2023) are high and low regularity tasks, respectively. Several measures could be used to formalize regularity. We advocate for the consistency score (C-score) of Jiang et al. (2020), that measures the expected accuracy of a model on held-out samples from the training set. Their precomputed CIFAR10 scores align with the regularity rankings of our tasks (see Fig. 13 in Jiang et al. (2020)). Adapted to $\mathcal{D}_c$, C-scores can be computed with $n = 0, 1, ...|\mathcal{D}_c|$ for instances $x$ and labels $y$, where $\mathbb{E}^r$ is empirical averaging over $r$ samples (Jiang et al., 2020):

$$\hat{C}_{\hat{\mathcal{D}}_c}(x, y) = \mathbb{E}_n[\hat{\mathbb{E}}^r_{D \overset{n}{\sim} \mathcal{D}_c}[\mathbb{P}(f(x; D \setminus \{(x, y)\}) = y)]]. \tag{1}$$

Failure along the regularity dimension for ETD is shown in Fig. 1 (d), where ETD succeeds at lower regularity tasks but fails at high regularity tasks. This is because information from regular samples, by design, reside in the generalization neurons of ETD, while low-regularity information (e.g.

example-specific peculiarities) is encoded in the memorization neurons that are dropped to induce forgetting. On the flip side, the Potion method excels at unlearning high regularity poisoned data, which is what it was designed for. However, it fails catastrophically at lower regularity unlearning tasks. This is because Potion is built on the assumption that the data that we wish to unlearn resides in a concentrated (small) set of locations in the network; a hypothesis that is likely to hold for high regularity data (e.g. some specific neurons may be disproportionately responsible for encoding the poison trigger) but is unlikely to hold for low regularity, where the corrupted data share no characteristics, making it unlikely for them to be stored in similar locations. Gradient Ascent (Fig. 1 (f)) also performs poorly at lower regularity tasks compared to high regularity tasks. We hypothesize that this is because the information of low regularity tasks is spread across a wider set of parameters, making it harder to erase without overly damaging the model utility using Gradient Ascent.

**Interplay between regularity and discovery.** As discussed before, traditional unlearning methods for full discovery are often based on fine-tuning on the retain set. For partial discovery, this is problematic as the retain set contains undiscovered corrupted data that are (re)introduced during fine-tuning (or retraining). However, interestingly, the degree of this (re)introduction from the partial set of corrupted data that lives in the retain set is a function of the regularity, with highly regular corruptions leading to amplification of this reintroduction. This is because for regular data, the presence of a few instances in the retain set suffice to (re)introduce the general shared pattern. For instance, the association that the poison trigger should redirect to a specific output is one that can be learned from a few poisoned examples, allowing the model to then predict the same output for other poisoned data that have the same trigger, without those being in the retain set, due to generalization. On the other hand, contamination of the retain set by low regularity corrupted data will lead to learning (or reinforcing) incorrect labels on that data but the damage won't spread beyond those specific examples. This is why, for partial discovery, retraining on $\mathcal{D}_r$ (Fig. 1(b)) suffers more pronounced and sudden drops the higher the regularity.

**Experimental setup.** We evaluate tasks of low, medium, and high regularity with ten discovery rates for each task (10%-100%). Low regularity is represented by random label swaps of $n$ samples to an incorrect label (Maini et al., 2023). In this setting, there is no regularity linking the samples or the new labels together. Medium regularity uses the inter-class confusion setup of Goel et al. (2022) where $n$ samples of two classes are swapped with each other. The highest regularity task uses a poison trigger to redirect to class 0 as defined by Goel et al. (2024). We measure the accuracy of $\theta_u$ on two types of data: (i) corrupted data (using the clean labels to compute accuracy), and (ii) non-corrupted data. The former is used to measure "healing", i.e. whether the label prediction has been "redirected" to the correct label, while the latter is used to measure "utility", i.e. ability to predict correctly on non-corrupted data. We perform experiments on an A100 (40GB) GPU with ResNet-9 (He et al., 2016) and Vision Transformer (ViT) (Dosovitskiy et al., 2021) models, stochastic gradient descent (SGD) and Adam (Kingma & Ba, 2014) optimizers, and CIFAR10 (Krizhevsky et al., 2009) and Street View House Numbers (SVHN) (Netzer et al., 2011) to show generalizability across models, optimizers and datasets. **GitHub: https://github.com/google-deepmind/rem**

## 5 REDIRECTION FOR ERASING MEMORY (REM)

We now introduce our core contribution, *Redirection for Erasing Memory (REM)*, the first unlearning method that is performant across our 2D space spanning different regularities and discovery rates, without requiring specialized hyperparameter tuning for each region. The design of REM is based on the lessons discussed in the previous section: (i) In the setting of partial discovery, finetuning on $\mathcal{D}_r$ causes reinforcing or reintroducing corruption (especially for high-regularity corruptions), but not using $\mathcal{D}_r$ at all would lead to low model utility; (ii) ETD can eliminate the effect of low-regularity corruptions by simply dropping out the memorization partition, but this operation will not remove the effect of high-regularity corruptions, as those are encoded in the generalization partition.

**Overview.** REM's key innovations are as follows. First, instead of avoiding the use of $\mathcal{D}_r$ (which would lead to low utility), REM accepts that undiscovered corrupted samples in $\mathcal{D}_r$ will unavoidably re-introduce corruptions but it employs a novel mechanism to redirect them to a dedicated part of the model at unlearning time that can then be dropped or deactivated. Second, the above is achieved through the design of a novel unlearning objective using a new loss and masking strategy.

Figure 2: REM performs the following steps: (i) **Expand** the network with randomly-initialized parameters $\theta_{o_2}$; (ii) **Remove** the corruptions out of $\theta_{o_1}$ with a SOTA unlearning algorithm on $\theta_{o_1}$ that does not use $\mathcal{D}_r$, avoiding reintroduction in $\theta_{o_1}$, but at the expense of utility; (iii) **Repair** utility by fine-tuning $\theta_{o_1}$ with $\mathcal{D}_{tr}$, using a novel **Redirection** strategy to steer any reintroduction of corruptions caused by the inclusion of $\mathcal{D}_r$ to the add-on parameters $\theta_{o_2}$; (iv) **Drop** out $\theta_{o_2}$.

Let $\theta_{o_1} = \mathcal{A}(\mathcal{D}_{tr})$ denote the parameters of the *original model*, trained using $\mathcal{A}$ on $\mathcal{D}_{tr}$. Unlike ETD, REM's choice of $\mathcal{A}$ can be any standard training procedure. REM then applies a post-processing $\mathcal{U}$ on top of $\theta_{o_1}$ to unlearn corrupted data by redirecting the effect of corruptions to newly-added parameters $\theta_{o_2}$ that are then dropped. We emphasize that, while this design draws inspiration from ETD, it is a fundamentally distinct methodology: while ETD attempts *at training time* to partition the network into generalization and memorization, REM attempts at *post-processing time* to partition the network into parameters that are unaffected by the corruptions and ones that capture all corruptions. Furthermore, REM adopts a unique masking strategy to capture high regularity corruptions that cannot be captured by ETD. We detail each REM step from Fig. 2 below and in Alg 1.

**Expand**. As a first step, REM expands the network with additional capacity, initialized randomly. These new parameters, denoted as $\theta_{o_2}$, can be seen as a placeholder at initialization, to which REM will attempt to redirect the corrupted data, as discussed in a later step. While REM allows complete freedom in the architectural design of $\theta_{o_1}$ and $\theta_{o_2}$, we adopt a simple approach here that enables a direct comparison with ETD: we expand the number of channels in each convolutional layer. The structure and capacity of each of $\theta_{o_1}$ and $\theta_{o_2}$ correspond directly to that of ETD's generalization and memorization parts. However in REM, $\theta_{o_1}$ is obtained by standard training (so it doesn't capture only "generalization"), while $\theta_{o_2}$ is a randomly-initialized expansion added at postprocessing time.

**Remove corruptions from** $\theta_{o_1}$ (step 2 of the pseudocode). We achieve this via Negative Preference Optimization (NPO) (Zhang et al., 2024a), a SOTA unlearning method in the NLP domain which we translate to our classification problem. NPO is chosen over Potion and Gradient Ascent, the other UL methods available that do not utilize $\mathcal{D}_r$, because Potion destroys model utility to an unrecoverable level at lower regularity tasks (see Fig. 1 (c)) and NPO has demonstrated better performance restoring or healing knowledge, as opposed to removing knowledge in the NLP domain (Rezaei et al., 2025), which is closely related to our goal of healing corrupted data. As research progresses, better UL methods that do not utilize $\mathcal{D}_r$ can replace NPO to boost REM performance further. NPO, adapted to classification by using the cross-entropy loss $\mathcal{L}_{CE} = -\sum_{c=1}^{C} y_c \log(\hat{y}_c)$, is shown as $\mathcal{L}_{\text{remove}_{\theta_{o_1}}}$ in Eq. 2 and used in step 2 of Alg. 1. Note that independence from the retain set during this step is essential to avoid reintroduction of corruptions in $\theta_{o_1}$. The downside of this is utility degradation which we address in the next step. The stop condition in Alg. 1 is adapted from Schoepf et al. (2024b), which shows that unlearning does not take effect gradually but suddenly as forgetting occurs. The parameter $\gamma$, as shown in Schoepf et al. (2024b), is not highly sensitive and should be chosen higher than random chance but lower than model utility.

**Repair utility and redirect corruptions into** $\theta_{o_2}$ **instead of relearning in** $\theta_{o_1}$. At this point, we have removed the effect of corruptions from $\theta_{o_1}$ at the expense of utility, via having unlearned on $\theta_{o_1}$ using an algorithm that doesn't use $\mathcal{D}_r$. We now aim to restore utility using $\mathcal{D}_{tr}$. This operation is risky since, for partial discovery, $\mathcal{D}_{tr}$ contains undiscovered corrupted data. We therefore design a loss function for this step that can benefit from the clean data of $\mathcal{D}_{tr}$ while attempting to redirect the corrupted data of $\mathcal{D}_c$ into $\theta_{o_2}$, which will then be discarded.

To that end, we design a masking strategy for $\theta_{o_2}$ analogously to ETD's masking strategy in its memorization partition. In particular, every example in $\mathcal{D}_{tr}$ will receive its own randomly determined mask in $\theta_{o_2}$. All examples in $\mathcal{D}_f$ are assigned the same (randomly-determined) mask as each

---

**Algorithm 1** Redirection for Erasing Memory (REM)

---

**Require:** Training dataset $\mathcal{D}_{\text{tr}}$ and forget set $\mathcal{D}_{\text{f}} \subset \mathcal{D}_{\text{tr}}$, Learning rate $\epsilon$, total epochs *max epochs*
**Require:** $\mathcal{D}_{\text{f}}$ removal stop condition: $\text{Acc}(\mathcal{D}_{\text{f}}) < \gamma$
  **Step 1:** Add additional capacity $\theta_{o2}$ to the existing model $\theta_{o1}$ for redirection
  **while** *current epoch $\leq$ max epochs* **do**
    **Step 2: Remove $\mathcal{D}_{\text{f}}$ from the $\theta_{o1}$ part of model**
      **while** $\text{Acc}(\mathcal{D}_{\text{f}}) > \gamma$ **do**
        2.1 Compute loss $\mathcal{L}_{\text{remove}_{\theta_{o1}}}(\mathcal{D}_{\text{f}})$ using only the $\theta_{o1}$ part of the model (see Eq. 2)
        2.2 Update $\theta_{o1}$ model part with loss $\mathcal{L}_{\text{step2}} = -\mathcal{L}_{\text{remove}_{\theta_{o1}}}$
    **end while**
    **Step 3: $\theta_{o1}$ model utility repair and $\mathcal{D}_{\text{c}}$ redirection into $\theta_{o2}$**
      3.1 Compute loss $\mathcal{L}_{\text{redirect}}$ using the $\theta_{o1} \cup \theta_{o2}$ model with $\mathcal{D}_{\text{tr}}$, where all $\mathcal{D}_{\text{f}} \subset \mathcal{D}_{\text{tr}}$ samples
         are assigned the same mask for redirection in $\theta_{o2}$; remaining samples use random masks
      3.2 Compute $\mathcal{L}_{\text{remove}_{\theta_{o1}}}(\mathcal{D}_{\text{f}})$ using only the $\theta_{o1}$ part of the model (see $\mathcal{L}_{\text{remove}_{\theta_{o1}}}$ in Eq. 2)
      3.3 Update model with the combined loss $\mathcal{L}_{\text{step3}} = \mathcal{L}_{\text{redirect}_{\theta_{o1} \cup \theta_{o2}}} - \mathcal{L}_{\text{remove}_{\theta_{o1}}}$ shown in Eq. 2
  **end while**
  **Step 4:** Discard additional capacity $\theta_{o2}$ to keep only $\theta_{o1}$ as the unlearned model.

---

other. Then, directly analogously to ETD, all examples pass through all of $\theta_{o_1}$ but each example will only pass through its assigned path in $\theta_{o_2}$ based on its mask. The intent is that forcing all forget set examples to share the same mask will cause the resulting path in $\theta_{o_2}$ to strongly learn the corruptions, turning into a "channel" for corrupted data. Utilizing the same mask for all corrupted samples intuitively seems like a problem, as the model may learn generalized knowledge in the added capacity when a neuron is present in multiple masks of samples that share generalizable knowledge. Specifically, when the added capacity is dropped, this generalized knowledge will be lost and model utility drops. This can indeed be true if the masking strategy is active from the start of training. However, in the post-hoc unlearning setting of REM, the model is already trained. Existing generalization will not switch over to the added capacity mask, as the path of least resistance is to reuse the already present neurons in the model that contain this information. We therefore first need to remove the existing information from these neurons (the Remove step of REM) to be able to redirect them. Because we had already unlearned corrupted data from $\theta_{o_1}$ in the previous step (a property which we reinforce further by adding NPO to the objective of this step), we are able to direct the corrupted data of the forget set to this channel in $\theta_{o_2}$.

At the same time as the redirection of corruptions, clean retain set examples can repair utility through fine-tuning $\theta_{o_1}$ which is kept clean from corruptions due to the previous removing step and the continued application of NPO on $\theta_{o_1}$ (Alg. 1 step 3.2). The combined loss (Alg. 1 step 3.3) is shown with $\sigma$ as the sigmoid function:

$$\mathcal{L}_{\text{step3}} = \underbrace{\frac{2}{\beta}\mathbb{E}\log\sigma\left(-\beta\log\left(\frac{\mathcal{L}_{\text{CE}_{\theta_{o_1} \cup \theta_{o_2}}}(\mathcal{D}_{\text{tr}})}{\mathcal{L}_{\text{CE}_{ref}}(\mathcal{D}_{\text{tr}})}\right)\right)}_{\mathcal{L}_{\text{redirect}_{\theta_{o_1} \cup \theta_{o_2}}}} - \underbrace{\frac{2}{\beta}\mathbb{E}\log\sigma\left(-\beta\log\left(\frac{\mathcal{L}_{\text{CE}_{\theta_{o1}}}(\mathcal{D}_{\text{f}})}{\mathcal{L}_{\text{CE}_{ref}}(\mathcal{D}_{\text{f}})}\right)\right)}_{\mathcal{L}_{\text{remove}_{\theta_{o_1}}}}. \quad (2)$$

The reference models ("*ref*") correspond to the initialization of the respective weights (both $\theta_{o_1}$ and $\theta_{o_2}$ in the first term; only $\theta_{o_1}$ in the second). This formulation is similar to Direct Preference Optimization (DPO) (Rafailov et al., 2023) with the key difference that we perform each part on a different model (i.e. a different set of parameters active and an example-dependent forward pass).

**REM on ETD**. While it's not necessary, REM can be applied over an ETD-trained model. In that case, we don't need to expand the network further; we directly use ETD's memorization partition for redirecting the corruptions, and ETD's existing masks for the examples in $\mathcal{D}_{\text{r}}$ (assigning a new one for examples in $\mathcal{D}_{\text{f}}$). All other steps of REM then proceed as usual. We evaluate this variation empirically too, finding that it boosts performance on low regularity and low discovery over plain REM, but this comes at a cost of some utility loss caused by the ETD training scheme (see appendix Fig. 8 for a comparison of the utility of pretrained models with and without ETD at different capacities).

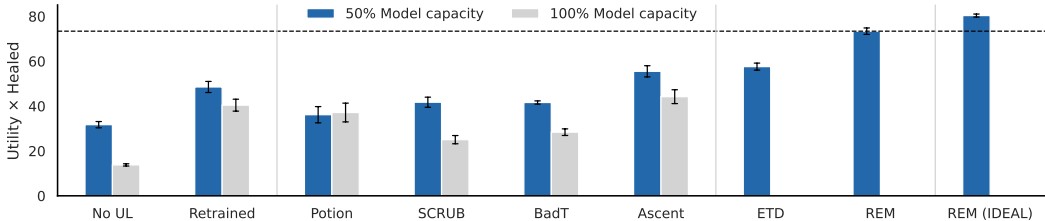

Figure 3: Comparison of UL methods on two model capacity levels using ResNet-9 & CIFAR10 with 1000 corrupted samples, three regularity levels and 10 discovery rates (10%-100%). REM (IDEAL) provides and upper limit with perfect knowledge of manipulated samples for mask assignment. ETD and REM are not reported for 100% capacity as reserve capacity is needed for $\theta_{o_2}$ of the model. Error bars reflect $\pm 1$ SEM.

Table 1: Aggregate results of methods on CIFAR10 with ResNet-9 and ViT with SVHN (50% capacity each) and 1000 corrupted samples with 10 discovery rate levels (10%-100%) and 3 regularities. Presented in descending order of Utility $\times$ Healed (ResNet-9). Potion ViT experiment failed due to OOM (Out of memory) with an A100 (40GB VRAM). Error reflects $\pm 1$ SEM.

| ETD | UL type | ResNet-9 | | | ViT |
| --- | --- | --- | --- | --- | --- |
| | | Healed | Utility | Utility×Healed | Utility×Healed |
| | REM | $81.16 \pm 1.62$ | $90.54 \pm 0.15$ | $\mathbf{73.40 \pm 1.43}$ | $\mathbf{73.27 \pm 0.32}$ |
| ✓ | REM | $\mathbf{83.26 \pm 0.92}$ | $88.05 \pm 0.18$ | $73.19 \pm 0.72$ | $72.80 \pm 0.36$ |
| ✓ | NPO | $77.50 \pm 1.53$ | $86.99 \pm 0.24$ | $67.10 \pm 1.17$ | $66.31 \pm 0.86$ |
| ✓ | Ascent | $76.59 \pm 1.44$ | $86.40 \pm 0.35$ | $65.82 \pm 1.08$ | $67.70 \pm 0.90$ |
| ✓ | SCRUB | $66.95 \pm 2.82$ | $89.45 \pm 0.14$ | $59.85 \pm 2.50$ | $55.03 \pm 3.24$ |
| ✓ | BadT | $66.24 \pm 1.89$ | $88.13 \pm 0.16$ | $58.32 \pm 1.63$ | $52.50 \pm 3.13$ |
| | NPO | $64.84 \pm 2.91$ | $87.59 \pm 0.36$ | $56.40 \pm 2.50$ | $65.62 \pm 0.88$ |
| | Ascent | $63.98 \pm 2.89$ | $86.97 \pm 0.29$ | $55.50 \pm 2.49$ | $67.14 \pm 0.89$ |
| ✓ | Potion | $62.74 \pm 3.73$ | $62.86 \pm 3.62$ | $51.30 \pm 3.64$ | OOM |
| ✓ | No UL | $56.97 \pm 3.15$ | $88.00 \pm 0.14$ | $50.11 \pm 2.75$ | $51.41 \pm 3.33$ |
| | Retrained | $53.61 \pm 2.73$ | $90.46 \pm 0.14$ | $48.52 \pm 2.47$ | $56.36 \pm 3.07$ |
| | SCRUB | $47.09 \pm 2.58$ | $\mathbf{90.71 \pm 0.15}$ | $42.69 \pm 2.33$ | $55.29 \pm 3.22$ |
| | BadT | $46.17 \pm 0.84$ | $90.23 \pm 0.18$ | $41.60 \pm 0.72$ | $52.40 \pm 3.08$ |
| | Potion | $49.39 \pm 3.61$ | $53.06 \pm 3.30$ | $36.16 \pm 3.62$ | OOM |
| | No UL | $35.33 \pm 1.66$ | $89.94 \pm 0.17$ | $31.72 \pm 1.46$ | $51.02 \pm 3.29$ |

## 6 RESULTS & DISCUSSION

REM represents a leap in coverage of the 2D space compared to previous methods as seen in Fig 1, setting a new SOTA. Unlike prior methods, REM does not exhibit a sudden breakdown along any of the framework dimensions and therefore presents itself as a safe choice for use in practice. All REM results use the same hyperparameters, showing generalizability across models, datasets and tasks.

**Metrics.** "Healed" computes the clean (i.e. uncorrupted label) accuracy on the corrupted $\mathcal{D}_c$ inputs. Utility computes the accuracy on a clean test set without any corruptions. Utility $\times$ Healed provides a multiplicative score, where higher is better, to ensure that the method not only removes the corrupted data but also retains model utility, analogous to Zhao et al. (2024). Figs. 1 (b-g) are shaded using this aggregate metric. Reported errors reflect $\pm 1$ standard error of the mean (SEM).

**Method comparison.** Fig. 1 shows that REM is the only method that performs strongly on all regions of our 2D space. We additionally provide aggregate numerical scores in Fig. 3, that are computed as the average of the Utility $\times$ Healed metric across runs for all discovery rates and regularities. We make the following observations. For all methods, the score increase for the 50% capacity model over the 100% model is due to less memorization of lower regularity corruptions due

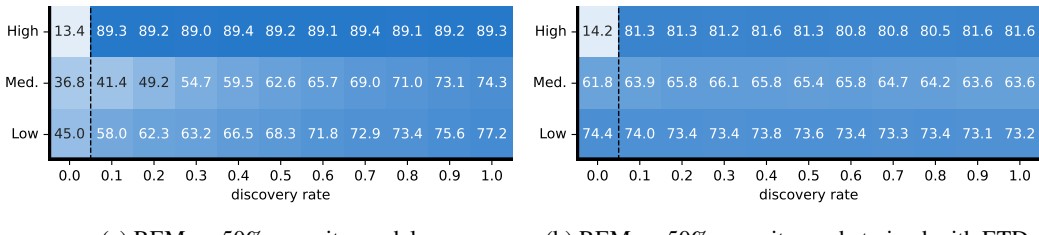

(a) REM on 50% capacity model    (b) REM on 50% capacity mode trained with ETD

Figure 4: Comparison of REM applied to a model trained with/without ETD. The performance of the model before applying REM is shown in the 0.0 discovery column. (b) shows that ETD provides an uplift in lower discovery rates for lower regularity tasks (y-axis) but comes at the cost of overall model utility (see Tab. 1) which harms the higher regularity and higher discovery rate performance.

to less overparameterization. We observe that ETD significantly outperforms the approach of simply training a smaller original model ("no UL" in Fig. 3). In fact, we find that ETD alone is a strong baseline that was not previously considered. It outperforms prior unlearning methods in terms of the aggregate scores, due to the effectiveness of memorization mitigation on lower regularity tasks. Further, as also discussed by Goel et al. (2024), retraining a model from scratch is not the gold standard for partial discovery tasks as the undiscovered corruption is reintroduced into the model, leading to poor performance. Potion does not perform well on this aggregate metric due to failure on the lower regularity tasks. A surprising discovery is that the simple baseline of Gradient Ascent beats many other methods on our 2D space in terms of aggregate score. This is a surprising as this baseline was believed to be weak and was even omitted from prior comparisons (Goel et al., 2024). However, we note that Ascent fails across entire slices of the 2D space as shown in Fig. 1, which is hidden in the aggregation but may be problematic in practice. REM outperforms all prior metrics and even comes close to REM (IDEAL), an upper bound that has knowledge of all corrupted data when forming masks at unlearning time.

**Unlearning on an ETD model.** Tab. 1 shows that REM is the best performing method when used either with or without ETD training. The choice of whether to use ETD pretraining leads to a trade-off, as shown in Fig. 4: ETD improves performance on the lower regularity tasks but its training procedure causes a model utility deficit as the model used during training differs from the model at inference time. Other unlearning methods show an increased combined score of Utility $\times$ Healed when combined with ETD but do not come close to REM. This is due to the shortcomings of the last used method affecting the results. For example, (i) in SCRUB, the reintroduction of the poison trigger in partial discovery settings causes failure in high regularity tasks, (ii) Potion causes significant model utility damage in the lower regularity settings negating the benefit of ETD.

**Prevention versus postprocessing.** As shown in Fig. 3, less overparameterization (smaller capacity) is effective in partly mitigating memorization of corruptions during training. We report extreme cases of this in the appendix (see Fig. 8), showing that by restricting model capacity to very low levels, no space for memorization remains, effectively solving lower regularity tasks but at the high cost of model utility. Therefore other mitigation actions during training, such as ETD for instance, should be preferred over extreme capacity restriction. Based on these insights, we argue that high regularity corruptions are the hardest to address during training. We suggest that this also applies to other domains such as NLP, where verbatim memorization can be addressed in simple ways such as with the Goldfish loss (Hans et al., 2024) but concepts (high regularity) are difficult to mitigate.

**Ablations.** Tab. 2 shows that redirection of manipulated samples is essential for REM performance. Other aspects of REM are more nuanced. ETD comes with the trade-off of more healing at the cost of utility. Step 3.2 in Alg. 1 increases the healing performance at higher discovery rates as this allows for better redirection by preventing reintroduction into the $\theta_{o_1}$ part of the model. When trained with ETD this difference vanishes as it is mainly observed in lower regularity tasks (Appendix Fig. 7).

**Different dataset/architecture/optimizer.** ViT (Adam, SVHN) results in Tab. 1 are in line with the ResNet-9 (SGD, CIFAR10) results with the key difference that the ViT model shows better mitigation against outliers / low regularity corruptions. This increases scores as the No ULs for medium and low regularity tasks are higher than in the ResNet-9 setting.

Table 2: REM ablation using CIFAR10 with ResNet-9 analogous to Tab. 1. Redirection (3.1) using $\mathcal{D}_{\text{tr}}$ instead of just finetuning on $\mathcal{D}_{\text{r}}$ is essential while step 3.2 only matters when the model is not trained with ETD. No use of 3.1 & 3.2 simplifies to NPO, as $\theta_{o_2}$ is unused. Error reflects $\pm 1$ SEM.

| Step 3.1 | Step 3.2 | ETD | Healed | Utility | Utility $\times$ Healed |
|---|---|---|---|---|---|
| ✓ | | ✓ | **83.70 $\pm$ 0.89** | 88.16 $\pm$ 0.16 | **73.69 $\pm$ 0.69** |
| ✓ | ✓ | | 81.16 $\pm$ 1.62 | 90.54 $\pm$ 0.15 | 73.40 $\pm$ 1.43 |
| ✓ | ✓ | ✓ | 83.26 $\pm$ 0.92 | 88.05 $\pm$ 0.18 | 73.19 $\pm$ 0.72 |
| ✓ | | | 78.94 $\pm$ 1.67 | **90.55 $\pm$ 0.15** | 71.38 $\pm$ 1.47 |
| | ✓ | | 77.31 $\pm$ 2.07 | 90.39 $\pm$ 0.14 | 69.82 $\pm$ 1.85 |
| | ✓ | ✓ | 78.83 $\pm$ 1.45 | 87.17 $\pm$ 0.21 | 68.68 $\pm$ 1.26 |

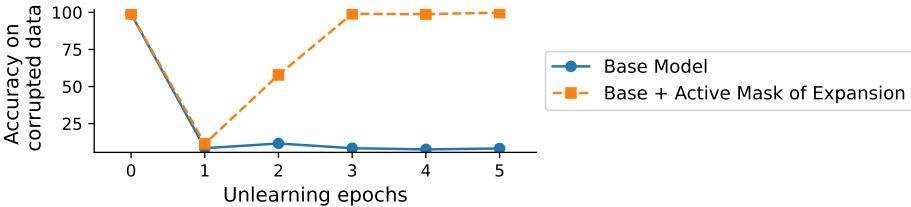

Figure 5: Accuracy on corrupted data for the base model (i.e. the existing trained model) and the base model plus the additional active parameters from the added mask. Epoch 0 is the starting point where no unlearning has taken place. As unlearning takes place, the corruptions are redirected into the additional capacity of the model and removed from the base model. This enables the base model to be free of corruptions by dropping the added capacity after unlearning.

**Redirection of corruptions.** We show the effectiveness of our redirection strategy using ResNet9, 50% discovery rate, and the poison trigger task over 5 unlearning epochs. As shown in Fig. 5, the base model starts off with 99.0% accuracy on the corrupted data (i.e. nearly every sample adversely affected). During unlearning, the base model (i.e. with no added parameters active) accuracy on corrupted data drops to around 10%, which is random chance with 10 classes. This is made possible because at the same time, the corruptions were rerouted into the newly added parameter partition, as shown by the accuracy on the corrupted data for the base model plus the active parameters of the expansion. These results clearly show that the corrupted examples are indeed rerouted, enabling unlearning without the problem of reintroduction that makes prior methods fail.

## 7 DISCUSSION AND CONCLUSION

We presented a taxonomy for corrupted data unlearning tasks along two dimensions: *discovery rate* and *regularity*, and showed that no prior algorithm succeeds across the entire space. We then proposed REM, the first unlearning method that performs strongly across this 2D space of unlearning tasks. REM redirects corrupted data to a dedicated model part that is dropped or deactivated after unlearning to remove the influence of corrupted data from the model. A limitation of REM is that, as in ETD, its masks are binary. Future work could consider softer masking strategies that may allow for better self-organization within $\theta_{o_2}$, leading to masks of (even undiscovered) corrupted data having greater overlap with one another. Such better masking strategies may lead to bridging the performance gap with REM (IDEAL) in Fig 3. However, in its current version, REM already makes important strides forward as the first universal method that is strong across our task space. We believe that our 2D framework is an important step forward in better understanding when different families of unlearning algorithms fail, offering useful tools and vocabulary for explaining their behaviours. We hope that future work expands our framework through identifying other key dimensions, builds upon it, for instance by translating its tasks to other modalities, and leverages it for investigating the behaviours of algorithms for other types of unlearning applications.

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

## 8    REPRODUCIBILITY STATEMENT (OPTIONAL)

We provide all necessary information for reproduction in our paper. The codebase of Goel et al. (2024) for the poison trigger and interclass confusions experiments is clearly highlighted in the main paper (*Experimental Setup*), along with the the random label swap experiments from Maini et al. (2023) and their implementation of ETD. The code for each of these can be found in the original repositories linked in the respective papers (Goel et al., 2022; Maini et al., 2023). Paired with our description of REM in Alg. 1, the detailed loss function in eq. 2, all necessary hyper-paremeters/model setups in Tab. 3, and the details given in *Experimental Setup*, the paper is fully reproducible. The GitHub repository is linked in the experimental setup paragraph.

## A  APPENDIX

### A.1  PRIOR WORK MAPPING IN OUR 2D SPACE

The following works are representative for the highlighted areas in Fig. 1 (a).

A  Maini et al. (2023)

B  Schoepf et al. (2024b)

C  Chundawat et al. (2023); Kurmanji et al. (2023)

D  Goel et al. (2024)

E  Zhao et al. (2024); Foster et al. (2024); Chundawat et al. (2023); Kurmanji et al. (2023)

F  Chundawat et al. (2023); Kurmanji et al. (2023)

### A.2  HEALING METRIC

The "Healing" results in this paper are reported on the corrupted data in the training data. We chose accuracy on the training data due to the the observations shown in Fig. 6. Due to the low variance on test data in lower regularity settings, the accuracy on corrupted data in the training data is significantly more informative. In the case of high regularity tasks, there is a near linear relationship between behaviour on the train and test data. For example, if the poison trigger is learned in the train data, it will also trigger in the test data and analogously if it is unlearned from the corrupted training data it will not be triggered on unseen test data. In the case of low regularity tasks, there is no connection between the training and the test set. Thus the low variance in the direction of the test accuracy axis. Any impacts of low regularity corruptions on overall model utility are already captured in the "Utility" accuracy that is reported on the test set to avoid falsification by overfitting on the training data.

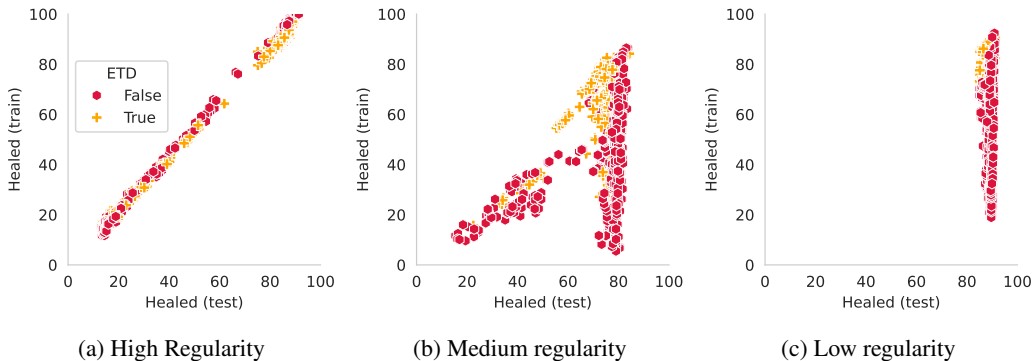

(a) High Regularity          (b) Medium regularity          (c) Low regularity

Figure 6: We report healing accuracy on the train and test set for all ResNet-9 experiments in the paper for all 10 discovery rates for each unlearning task (three regularity levels) and all UL methods. Results show that healing corrupted data is highly correlated between train and test data for corruptions with high regularity while showing little correlation for low regularity manipulations such as *Rand. Label Swap*. Models that were destroyed during unlearning (model utility below 80%) were removed.

### A.3  MODEL AND UL METHOD PARAMETERS

We perform unlearning with a limited hyperparameter sweep as the best values are already known for high and lower regularity tasks and varying discovery rates from Goel et al. (2024) (poison trigger and interclass confusion tasks). We report the selected hyperparameters in Tab. 3. The ViT architecture and setup for SVHN is adapted from Torzadehmahani et al. (2024), only changing the optimizer to Adam and adapting the learning rate and batch size accordingly. The ResNet-9 setup as well as the poison trigger and interclass confusion tasks are used from Goel et al. (2024). The random label unlearning problem is adapted from Maini et al. (2023). The REM parameter $\beta$

Table 3: (Hyper)parameter overview. The UL learning rate for ResNet-9 is the final learning rate of the SGD scheduler during training as set by Goel et al. (2024). We use the same 1/5 fraction from starting learning rate to unlearning learning rate for ViT.

| Method / Training | Hyperparameter | Values |
|---|---|---|
| ResNet-9 | Learning rate | 0.025 |
| ViT | Learning rate | 1e-4 |
| All architectures | Epochs | 40 |
| ResNet-9 | Batch size | 512 |
| ViT | Batch size | 2048 |
| ViT | Patch Size | 8 |
| ViT | Embedding Dimension | 512 |
| ViT | Transformer Depth | 4 |
| ViT | Attention Heads | 8 |
| ViT | MLP Hidden Dimension | 1536 |
| ViT | Head Dimension | 64 |
| ViT | Optimizer | Adam |
| ResNet-9 | Optimizer | SGD |
| ETD (ResNet) | Layer locations | After each ResNet-9 block |
| ETD (ViT) | Layer locations | Before/after FF in transformer blocks |
| All | Seeds | 0, 1, 2 |
| Hardware | GPU | A100 (40GB) |
| REM, Potion, NPO, Ascent | Threshold $\gamma$ | 0.2 |
| REM | $\beta$ | 1 |
| SCRUB | $\alpha$ | 0.001, 0.01, 0.1, 10 |
| All UL methods | UL learning rate | 0.005 (ResNet-9), 2e-5 (ViT) |
| All UL methods | Max. UL epochs | 10 (25% of training) |

is kept at 1 to avoid adding extra complexity to the method by introducing weighting.. Training data splits, choice of which sample IDs to manipulate etc. are taken 1:1 from Goel et al. (2024) to ensure unbiased selection. The hyperparameters for unlearning methods are taken from Goel et al. (2024) and Schoepf et al. (2024b). For ETD and REM each mask has 20% of the "memorization" partition active during training. The "memorization" partition size is the leftover capacity from "generalization" part to 100% capacity model. For a sensitivity analysis of "generalization" part and mask sizes please refer to Maini et al. (2023). From our experiments that are seeded with a global seed (affecting masks), we observe that REM is not sensitive to the random seed of the mask. Given the three seeds (0, 1, 2), we get the following healing and utility in the ResNet + CIFAR10 random label unlearning setting for 50% discovery rate (representative halfway point of discovery): Healing [0.772, 0.774, 0.752,], Utility [0.891, 0.888, 0.896]. Results reported for random label unlearning, as this scenario has the least regularity and highest randomness. Training time on one A100 (40GB) GPU for one seed for the reported results on ResNet-9 and ViT across training types, discovery rates, regularity levels, any UL method hyperparameters is ca. one week depending on the frequency of logging accuracy results.

Regarding generalisability across architectures, our experimental results show that REM works across architectures with Transformers and ResNet. The implementation in the ViT is agnostic to vision tasks as we place the REM expansions before/after the FF layers in each transformer block. This setup is applicable to transformer models across modalities but lies outside the scope of this paper. Ghosal et al. (2025) also shows that the ETD approach can be scaled to LLMs for memorization prevention during training (not an unlearning application), indicating that REM is highly likely to succeed at larger scales.

## A.4 REM ABLATION

Extending upon the ablations in Table 2, Fig. 7 shows that the barrier step (3.2 in Alg 1) adds (i) stability to the results without the performance jumps observed without the barrier step (ii) and leads

to better performance at most discovery rates and regularity levels. The few instances where REM without the barrier step outperforms the default REM indicate that adding the barrier to the loss causes some performance loss which is more pronounced at lower discovery rates when the loss is noisier due to the smaller sample size.

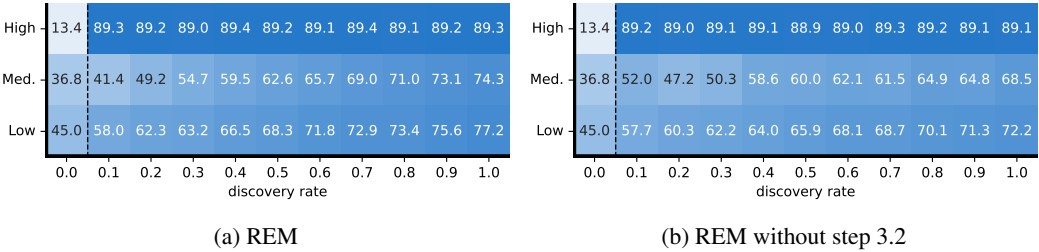

(a) REM                                    (b) REM without step 3.2

Figure 7: Comparison of REM variants for 50% capacity model on CIFAR10.

## A.5 MODEL CAPACITY AND ADDITIONAL ETD INSIGHTS

In Fig. 8 we observe that ETD may be approximated by models with smaller capacity that prevent memorization due to their smaller parameter count. However, in practice it is infeasible to search for a model cpacity that provides the perfect utility / memorization mitigation trade-off. Our experiments further show that both smaller models and ETD models fail to mitigate high regularity corruptions (see Fig. 8 (c)). This is expected, as high regularity corruptions are informally speaking no different than other concepts the model learns (e.g., concept of a stop sign vs concept of a poison trigger). We also perform an experiment where we apply our REM masking strategy of assigning each identified corrupted sample the same mask to capture higher regularity corruptions to the model training stage. By giving this combination of REM and ETD strategies full knowledge of all corrupted samples (information that is not available in practice a training time), we can show that high regularity concepts can be captured effectively as shown in Fig. 9. This is not an unlearning method but a demonstration of the effectiveness of our novel masking strategy to capture high regularity corruptions. Future work could investigate if leaving already identified corruptions in the training data can be beneficial when adopting this masking strategy from REM to create a "channel" during training that may capture additional undiscovered corruptions of similar nature.

## A.6 DETAILED RESULTS

We provide detailed results from our experiments in tables and heatmaps that reflect the 2D dimensions of our framework from Fig. 1 (a).

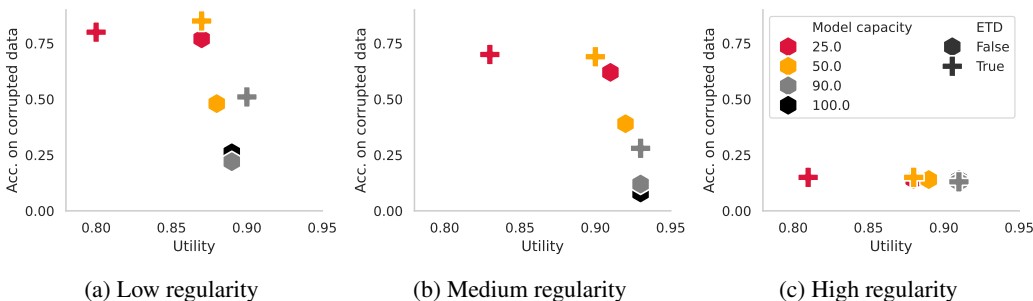

(a) Low regularity       (b) Medium regularity       (c) High regularity

Figure 8: We compare ETD against non-ETD models of the same capacity level, i.e. the parameter count at inference, where for ETD, only the generalization partition is active at inference time. The y-axis measures the clean-label accuracy on corrupted data where higher is better. We observe: (i) for low and medium regularity, lower-capacity models fit corrupted data less well compared to higher-capacity models, acting as a type of regularizer against learning corruptions in the first place. However, lower capacity models are generally associated with lower utility. (ii) in this region, ETD greatly improves at unlearning the corruptions over its non-ETD counterpart of comparable capacity, which however comes at a cost of utility, especially at lower capacities. (iii) For high regularity tasks, neither capacity restriction nor ETD can move the needle in terms of unlearning corruptions. The drastic utility drop at very low model capacity with ETD is likely due to the stark mismatch between the model parameters active during training (for which the loss function optimizes) and inference.

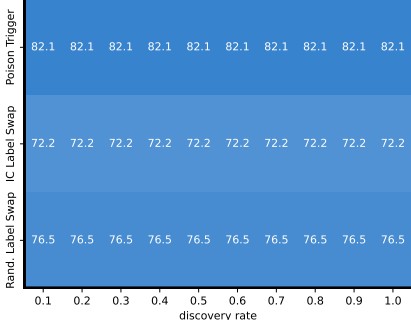

Figure 9: Model trained using the generalization / memorization split of ETD but with the masking strategy of REM and given full information about all corrupted samples.

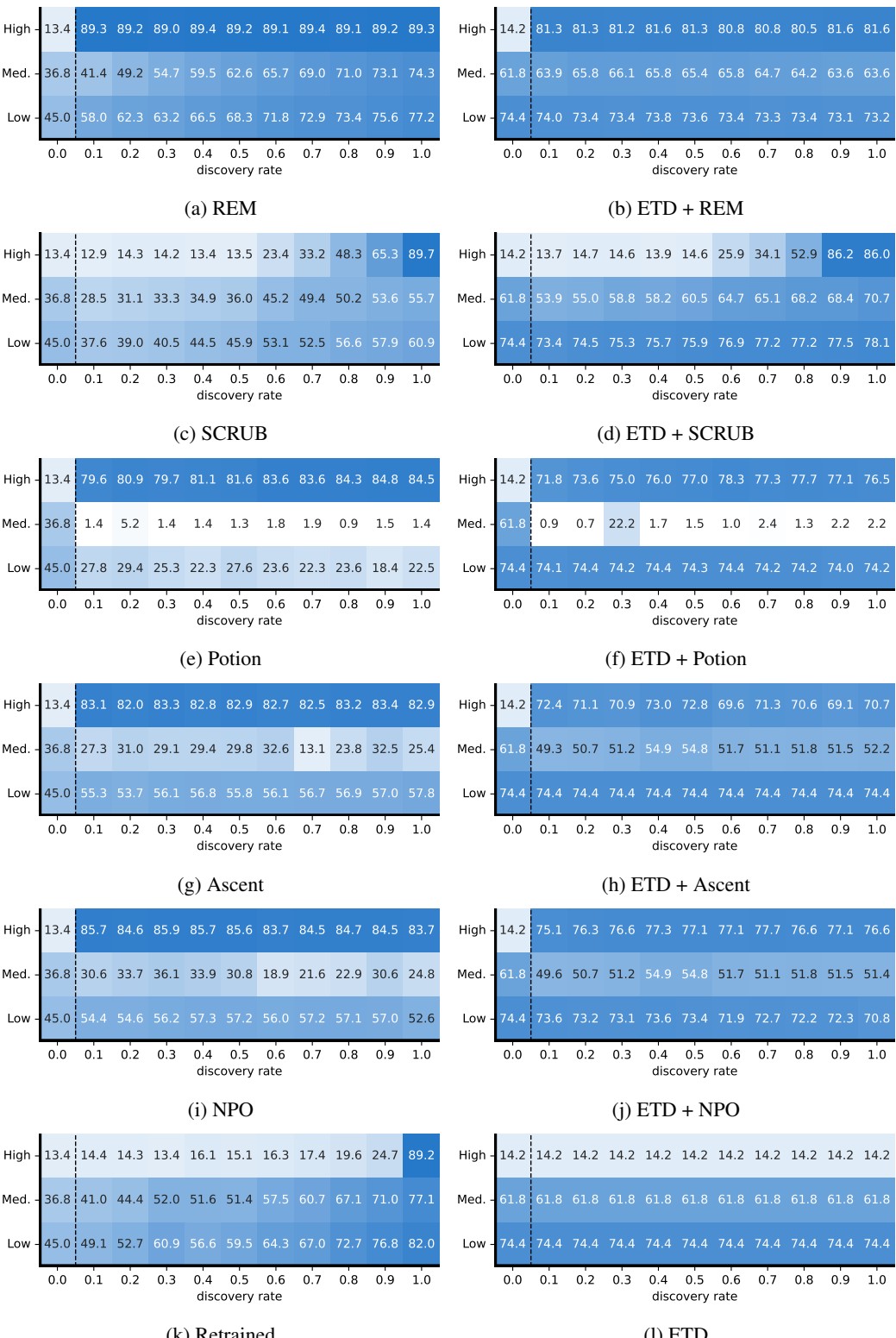

Figure 10: Healed × Utility heatmaps for ResNet-9 with CIFAR10 and a 50% capacity model unless otherwise specified. Left of dashed line is no unlearning (i.e. trained model before unlearning).

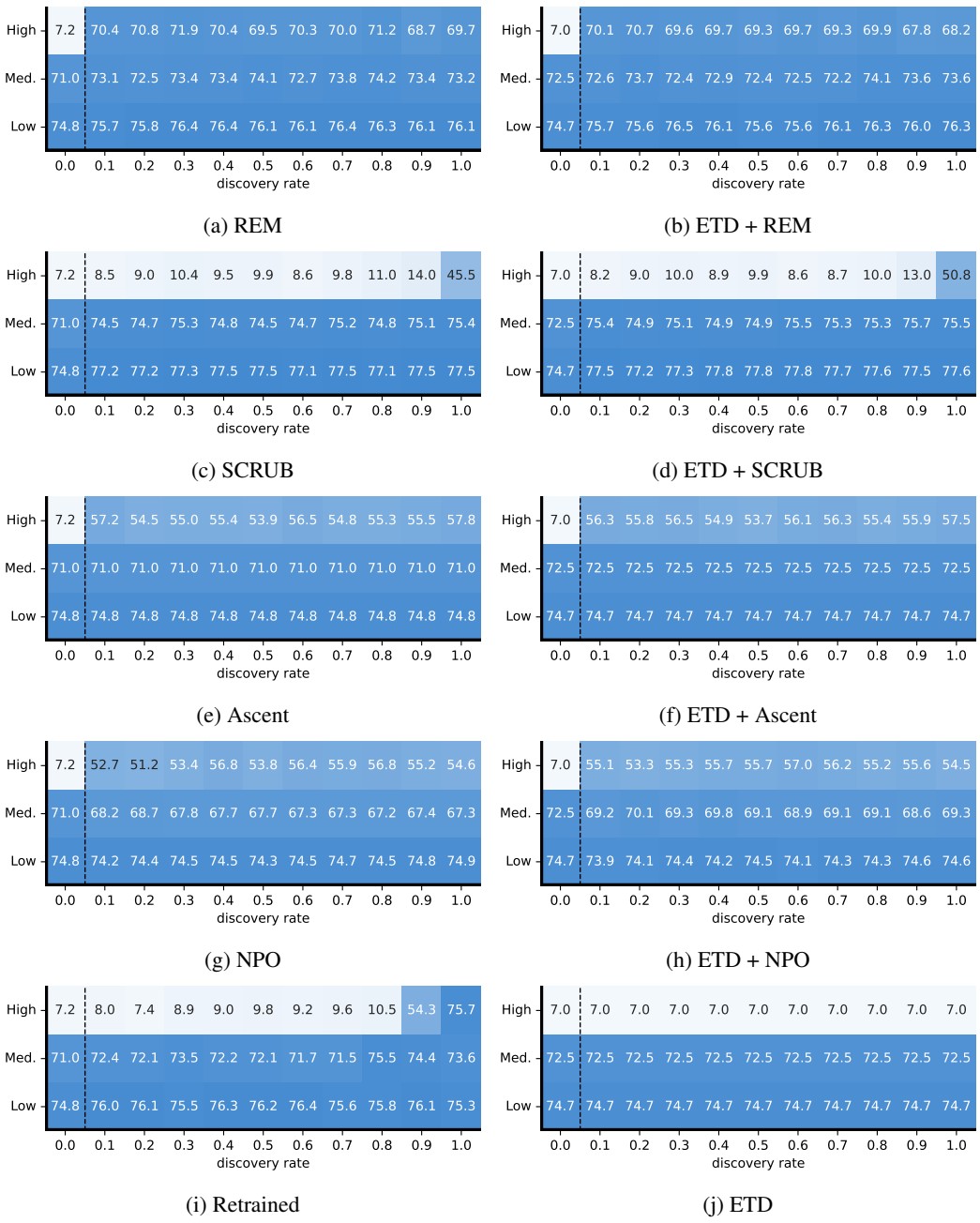

Figure 11: Healed × Utility heatmaps for ViT with SVHN and a 50% capacity model unless otherwise specified. Left of dashed line is no unlearning (i.e. trained model before unlearning).

Table 4: Aggregate results of methods on CIFAR10 with ResNet-9 (50% capacity each) and 1000 corrupted samples with 10 discovery rate levels (10%-100%) and 3 regularities. Presented in descending order of Utility × Healed. Error reflects ±1 SEM.

| Capacity | ETD | UL type | Healed | Utility | Utility × Healed |
|---|---|---|---|---|---|
| 50 |  | REM | 81.16 ± 1.62 | 90.54 ± 0.15 | **73.40 ± 1.43** |
| 50 | ✓ | REM | **83.26 ± 0.92** | 88.05 ± 0.18 | 73.19 ± 0.72 |
| 50 | ✓ | NPO | 77.50 ± 1.53 | 86.99 ± 0.24 | 67.10 ± 1.17 |
| 50 | ✓ | Ascent | 76.59 ± 1.44 | 86.40 ± 0.35 | 65.82 ± 1.08 |
| 50 | ✓ | SCRUB | 66.95 ± 2.82 | 89.45 ± 0.14 | 59.85 ± 2.50 |
| 100 |  | REM | 65.35 ± 3.28 | 91.33 ± 0.16 | 59.65 ± 2.99 |
| 50 | ✓ | BadT | 66.24 ± 1.89 | 88.13 ± 0.16 | 58.32 ± 1.63 |
| 50 |  | NPO | 64.84 ± 2.91 | 87.59 ± 0.36 | 56.40 ± 2.50 |
| 50 |  | Ascent | 63.98 ± 2.89 | 86.97 ± 0.29 | 55.50 ± 2.49 |
| 50 | ✓ | CF | 60.24 ± 3.07 | 89.83 ± 0.13 | 54.13 ± 2.75 |
| 50 | ✓ | Potion | 62.74 ± 3.73 | 62.86 ± 3.62 | 51.30 ± 3.64 |
| 50 | ✓ | No UL | 56.97 ± 3.15 | 88.00 ± 0.14 | 50.11 ± 2.75 |
| 50 |  | Retrained | 53.61 ± 2.73 | 90.46 ± 0.14 | 48.52 ± 2.47 |
| 100 |  | NPO | 54.54 ± 3.87 | 87.66 ± 0.37 | 47.20 ± 3.33 |
| 50 |  | CF | 49.08 ± 2.61 | 90.69 ± 0.15 | 44.51 ± 2.36 |
| 100 |  | Ascent | 51.98 ± 3.69 | 85.85 ± 0.54 | 44.20 ± 3.09 |
| 50 |  | SCRUB | 47.09 ± 2.58 | 90.71 ± 0.15 | 42.69 ± 2.33 |
| 50 |  | BadT | 46.17 ± 0.84 | 90.23 ± 0.18 | 41.60 ± 0.72 |
| 100 |  | Retrained | 44.13 ± 2.90 | 91.46 ± 0.13 | 40.41 ± 2.66 |
| 100 |  | Potion | 46.79 ± 4.27 | 55.78 ± 3.89 | 37.13 ± 4.19 |
| 50 |  | Potion | 49.39 ± 3.61 | 53.06 ± 3.30 | 36.16 ± 3.62 |
| 100 |  | CF | 38.82 ± 2.59 | **91.59 ± 0.14** | 35.53 ± 2.37 |
| 50 |  | No UL | 35.33 ± 1.66 | 89.94 ± 0.17 | 31.72 ± 1.46 |
| 100 |  | BadT | 31.18 ± 1.63 | 91.28 ± 0.16 | 28.37 ± 1.47 |
| 100 |  | SCRUB | 26.97 ± 2.05 | 91.55 ± 0.14 | 24.63 ± 1.86 |
| 100 |  | No UL | 15.24 ± 0.60 | 90.87 ± 0.15 | 13.78 ± 0.52 |

Table 5: Aggregate results of methods on SVHN with ViT (50% capacity each) and 1000 corrupted samples with 10 discovery rate levels (10%-100%) and 3 regularities. Presented in descending order of Utility × Healed. Values are to the best method for each discovery rate - regularity pairing (best by Utility × Healed). Potion OOM and thus not reported. Error reflects ±1 SEM. No UL and Ascent (best performing prior method) with 100% capacity are reported for reference to the 50% capacity models to show that the observations from ResNet experiments on capacity levels are applicable too.

| Capacity | ETD | UL type | Healed | Utility | Utility × Healed |
|---|---|---|---|---|---|
| 50 | | REM | **84.67 ± 0.38** | 86.53 ± 0.03 | **73.27 ± 0.32** |
| 50 | ✓ | REM | 84.17 ± 0.42 | 86.49 ± 0.03 | 72.80 ± 0.36 |
| 50 | ✓ | Ascent | 81.09 ± 0.74 | 83.19 ± 0.39 | 67.70 ± 0.90 |
| 50 | | Ascent | 80.28 ± 0.74 | 83.35 ± 0.38 | 67.14 ± 0.89 |
| 100 | | Ascent | 79.96 ± 1.16 | 82.99 ± 0.78 | 66.56 ± 1.47 |
| 50 | ✓ | NPO | 79.56 ± 0.71 | 83.08 ± 0.38 | 66.31 ± 0.86 |
| 50 | | NPO | 78.65 ± 0.78 | 83.18 ± 0.36 | 65.62 ± 0.88 |
| 50 | | Retrained | 65.33 ± 3.56 | 86.26 ± 0.03 | 56.36 ± 3.07 |
| 50 | | SCRUB | 63.36 ± 3.70 | 87.32 ± 0.02 | 55.29 ± 3.22 |
| 50 | ✓ | SCRUB | 63.03 ± 3.71 | **87.33 ± 0.02** | 55.03 ± 3.24 |
| 50 | ✓ | CF | 62.32 ± 3.70 | 87.06 ± 0.03 | 54.27 ± 3.22 |
| 50 | | CF | 62.14 ± 3.70 | 87.03 ± 0.03 | 54.07 ± 3.22 |
| 50 | ✓ | BadT | 61.20 ± 3.65 | 85.87 ± 0.02 | 52.50 ± 3.13 |
| 50 | | BadT | 61.01 ± 3.59 | 85.97 ± 0.02 | 52.40 ± 3.08 |
| 50 | ✓ | No UL | 59.94 ± 3.89 | 85.83 ± 0.02 | 51.41 ± 3.33 |
| 50 | | No UL | 59.44 ± 3.83 | 85.92 ± 0.02 | 51.02 ± 3.29 |
| 100 | | No UL | 59.30 ± 2.43 | 85.83 ± 0.56 | 50.94 ± 2.92 |

Table 6: Aggregate results of methods on Imagenette with ResNet (50% capacity each, no ETD pretraining) and 100 corrupted samples with 10 discovery rate levels (10%-100%) and 3 regularities. Presented in descending order of Utility × Healed. Values are to the best method for each discovery rate - regularity pairing (best by Utility × Healed). Error reflects ±1 SEM.

| UL type | Healed | Utility | Utility × Healed |
|---|---|---|---|
| REM | 76.03 ± 2.61 | 80.32 ± 0.28 | 60.98 ± 2.02 |
| SCRUB | 67.63 ± 3.63 | 80.66 ± 0.17 | 54.52 ± 2.89 |
| Retrained | 65.93 ± 3.04 | 78.17 ± 0.27 | 51.49 ± 2.32 |
| Potion | 56.40 ± 4.84 | 72.44 ± 2.50 | 42.61 ± 3.83 |
| BadT | 47.90 ± 3.18 | 79.64 ± 0.31 | 38.09 ± 2.50 |
| Ascent | 51.37 ± 2.67 | 75.38 ± 1.38 | 38.05 ± 1.66 |
| No UL | 31.00 ± 8.08 | 79.73 ± 1.16 | 24.69 ± 6.38 |

## A.7    ADDITIONAL EXPERIMENTS WITH DIFFERENT DATA, CORRUPTION COUNTS, AND CAPACITY LEVELS

We train the same ResNet as for CIFAR10 with the same learning rate etc. for the same 40 epochs on Imagenette[1] (see Tab. 6 & 7) but with 100 corruptions (to add a different size for variety). The new experiments again show REM as the top performing method, even with a 90/10 capacity split (Tab. 7). The gap between methods is smaller due to only 100 corruptions having lower influence via reintroduction of corruptions in partial discovery settings (i.e. methods fail later in the high regularity setting). Tables 8 and 9 show the same experiments as in the main body of the paper but with ETD/REM masks that are 1/4 the size of the main body experiments (20% vs 5%) to show that results hold across different sizes and do not vary greatly.

---

[1]https://github.com/fastai/imagenette

Table 7: Aggregate results of methods on Imagenette with ResNet (90% capacity each, with ETD pretraining) and 100 corrupted samples with 10 discovery rate levels (10%-100%) and 3 regularities. Presented in descending order of Utility × Healed. Values are to the best method for each discovery rate - regularity pairing (best by Utility × Healed). Error reflects ±1 SEM.

| UL type | Healed | Utility | Utility × Healed |
|---------|--------|---------|------------------|
| REM | $77.33 \pm 2.72$ | $79.51 \pm 0.21$ | $61.41 \pm 2.09$ |
| SCRUB | $68.83 \pm 3.90$ | $79.96 \pm 0.19$ | $55.02 \pm 3.07$ |
| Ascent | $66.97 \pm 2.01$ | $75.17 \pm 0.73$ | $50.21 \pm 1.44$ |
| Retrained | $64.67 \pm 2.93$ | $77.54 \pm 0.21$ | $50.14 \pm 2.26$ |
| Potion | $64.77 \pm 3.34$ | $69.90 \pm 1.64$ | $46.28 \pm 3.16$ |
| BadT | $55.57 \pm 2.49$ | $78.59 \pm 0.13$ | $43.62 \pm 1.92$ |
| No UL | $45.33 \pm 8.25$ | $78.50 \pm 0.44$ | $35.53 \pm 6.30$ |

Table 8: Aggregate results of methods using smaller masks (5% active vs 20% in main experiments) on CIFAR10 with ResNet-9 (50% capacity each) and 1000 corrupted samples with 10 discovery rate levels (10%-100%) and 3 regularities. Presented in descending order of Utility × Healed. Error reflects ±1 SEM.

| UL type | Healed | Utility | Utility × Healed |
|---------|--------|---------|------------------|
| REM | $\mathbf{85.37 \pm 1.77}$ | $89.56 \pm 0.26$ | $\mathbf{76.38 \pm 1.48}$ |
| Potion | $78.54 \pm 2.45$ | $85.84 \pm 0.68$ | $67.68 \pm 2.41$ |
| Ascent | $75.72 \pm 3.01$ | $88.75 \pm 0.47$ | $66.81 \pm 2.38$ |
| NPO | $75.88 \pm 3.25$ | $88.03 \pm 0.53$ | $66.33 \pm 2.54$ |
| BadT | $64.64 \pm 2.74$ | $89.28 \pm 0.27$ | $57.57 \pm 2.32$ |
| Retrained | $59.86 \pm 4.94$ | $88.79 \pm 0.24$ | $53.21 \pm 4.39$ |
| SCRUB | $59.04 \pm 4.92$ | $\mathbf{89.99 \pm 0.25}$ | $53.02 \pm 4.36$ |
| No UL | $51.87 \pm 19.89$ | $89.47 \pm 0.93$ | $46.31 \pm 17.54$ |

Table 9: Aggregate results of methods using smaller masks (5% active vs 20% in main experiments) on SVHN with ViT (50% capacity each) and 1000 corrupted samples with 10 discovery rate levels (10%-100%) and 3 regularities. Presented in descending order of Utility × Healed. Error reflects ±1 SEM.

| UL type | Healed | Utility | Utility × Healed |
|---------|--------|---------|------------------|
| REM | $\mathbf{81.19 \pm 1.16}$ | $85.38 \pm 0.10$ | $\mathbf{69.33 \pm 1.01}$ |
| Ascent | $77.47 \pm 1.80$ | $81.41 \pm 0.85$ | $63.49 \pm 2.03$ |
| SCRUB | $63.26 \pm 6.28$ | $\mathbf{86.21 \pm 0.09}$ | $54.61 \pm 5.43$ |
| BadT | $61.38 \pm 5.69$ | $83.96 \pm 0.14$ | $51.61 \pm 4.80$ |
| No UL | $59.90 \pm 24.29$ | $84.40 \pm 0.45$ | $50.64 \pm 20.55$ |
| NPO | $59.58 \pm 6.42$ | $84.40 \pm 0.12$ | $50.37 \pm 5.43$ |
| Retrained | $58.81 \pm 4.91$ | $76.32 \pm 0.24$ | $44.98 \pm 3.75$ |

## A.8  COMPUTE TIME AND MEMORY

We report unlearning time as relative numbers compared to initial training for the CIFAR10 runs with a 50% capacity model (average runtimes). Initial training 100%, Bad Teacher 24.7%, SCRUB 14.0%, REM 10.7%, Potion 9.1%, NPO 0.2%, Ascent 0.2%. A key consideration in our benchmark is that no prior method, not even retraining from scratch, can address the 2D space of unlearning tasks presented. This is different to privacy focused unlearning, where retraining from scratch is the gold standard. Therefore, being more efficient is a necessity in privacy focused unlearning - otherwise retraining is better. Given that this is not the case for corrupted data, time is not essential until multiple methods are able to address the 2D space (REM is fast nonetheless compared to training from scratch, SCRUB, etc. and only misses out to methods that do not perform any repair steps). Regarding memory requirements, for REM this depends on the size of $\theta_{o_2}$ to keep the expanded model in memory. Some methods such as Potion have higher peak memory usage due to expensive parameter importance computations (see OOM problem with ViT), while others are more efficient due to no $\theta_{o_2}$ as in Ascent, or are harder to compare due to multi-model student-teacher setups as in SCRUB and Bad Teacher.

## A.9  LIMITATIONS

We highlight the key limitations and assumptions in the main paper. On the unlearning side this is the assumption of having access to the full retain set at unlearning time as is common in the literature. We leave studies of more restrictive settings to future work. The main limitation of REM that harms performance is its masking strategy as highlighted in the paper. We also show the potential that can be unlocked by improved masking with REM (IDEAL) that uses information that is not available in practice to reflect perfect masking. Due to imperfect masking, REM will not perform to its maximum potential in practice, which we call out and provide inspiration for future work to address this. We also note that the removal step can be improved by using future advancements in unlearning methods that do not rely on a retain set to replace our implementation of NPO for this step. Depending on the used method for removal, the limitations of the chosen method will be inherited by REM. We show that REM is robust across tasks, models, optimizers and datasets. Prior UL literature has shown that unlearning methods show stable results across varying dataset sizes in high regularity tasks Schoepf et al. (2024b) as well as lower regularity tasks of the same nature (e.g. vision classifiers). REM's runtime is dependent on the chosen number of epochs and comparable to an epoch of normal training as step 2 of REM uses the much smaller forget set and is negligible in comparison to the backpropagation step on the full training data.

