# OpenReview forum: "Redirection for Erasing Memory (REM): Towards a universal unlearning method for corrupted data"
_ICLR.cc/2026/Conference — ICLR 2026 Poster_

### Official Review · Reviewer_UzFN · 2025-10-16

**Soundness:** 2
**Presentation:** 3
**Contribution:** 3
**Rating:** 4
**Confidence:** 5

**Summary:**

This paper addresses the challenge posed by the specialization of machine unlearning (UL) methods to particular tasks, which has complicated their systematic comparison and resulted in unpredictable failures in practice. To characterize diverse corrupted data unlearning tasks in vision classifiers, the authors propose a conceptual taxonomy described by two key dimensions: the **discovery rate** (the fraction of corrupted data known at the time of unlearning) and the **statistical regularity** of the corrupted data. Previous State-of-the-Art (SOTA) methods were typically designed for specific regions within this 2D space and are shown to fail predictably and catastrophically outside of these targeted areas. To overcome this limitation and provide a universal solution, the authors introduce **Redirection for Erasing Memory (REM)**. The core feature of REM involves redirecting corrupted data to dedicated neurons that are newly introduced to the model during the unlearning process. Subsequently, these dedicated neurons are discarded or deactivated to suppress the influence of the corrupted data. Experimental results demonstrate that REM is the first unlearning algorithm that performs strongly across the entire spectrum of tasks within this defined 2D space, setting a new SOTA and proving to be a safe choice for practical applications.

**Strengths:**

- Proposes a clear 2D taxonomy (discovery rate × statistical regularity) that explains why prior unlearning methods succeed only in narrow slices and fail predictably elsewhere.

- Introduces REM, a simple yet novel “redirection” mechanism that routes corrupted data into added neurons and then drops them—yielding strong performance across the entire task space.

- Backs claims with broad, rigorous benchmarks (CIFAR-10/SVHN; ResNet-9/ViT), showing SOTA scores and practical efficiency.

**Weaknesses:**

- The stated motivation is that existing unlearning methods sometimes fail unpredictably an that they lack a principled understanding of the causes.Yet the paper offers no deep causal analysis; it primarily introduces a taxonomy (but valuable).

- Step 2 states that the original model is updated using $L_{step2} (L_{remove})$. However, in Step 3 the model is updated using $L_{step3}$, where a minus sign is placed in front of $L_{remove}$. Even if Step 2 is indeed the unlearning phase, the current wording is ambiguous.

- The problem setting is well-motivated and realistic. That said, the assumption of full access to the entire training set is difficult to justify in practice. Although the authors defend this design choice as a means to preserve model utility, it would be better that exploring alternatives that operate with only a subset of the training data.

- The loss function notation seems inconsistent, $L$ and $\mathcal{L}$ are used confusingly.

**Questions:**

- Most prior unlearning research treats a fully retrained model as the gold-standard, by the definition of unlearning. However, this paper considers scenarios in which retraining is not a valid gold-standard. Under that premise, can this work still be considered, in a fundamental sense, a contribution to machine unlearning?

- I'm wondering whether the Utility$\times$Healed metric is essential. Would separate reporting convey the trade-off more transparently in ViT case?

- For LLM unlearnings, what are practical ways to capture or quantify "statistical regularity"?

---

> ### Author Response · Authors · 2025-11-19
>
> Dear reviewer UzFN, thank you for your feedback.
>
> ---
>
> Q1: We agree with the reviewer that this unlearning formulation is a little bit different than the classic formulation of unlearning motivated by privacy applications, but we do think that it still falls under the umbrella of unlearning; see Goel et al. for more discussion on this type of "corrective unlearning" definition. Note that retrain-from-scratch excluding all of the corrupted data (i.e. 100% discovery case) is still the oracle here, but unlike the standard problem, we can't run this oracle (even given sufficient compute), since we only have partial knowledge of the corrupted data. Generally, the field of unlearning is starting to split into privacy/sensitive data and corrupted data (data poisoning, noisy labels, …) streams, each with their distinct challenges. We are contributing to the latter, which is starting to receive more attention as publications on the feasibility and danger of model poisoning garner more attention (e.g. the recent Anthropic study that found that as few as 250 malicious documents can produce a "backdoor" vulnerability in a large language mode -> https://arxiv.org/abs/2510.07192).
>
> ---
>
> Q2: We provided the detailed results for ViT in the appendix (Table 5 and Fig. 10). We apologize if this was not apparent and will call this out more clearly in the main text. The granular findings are in line with the ResNet results, such as SCRUB failing in high regularity tasks as soon as the discovery rate drops below 100%, retrain set free methods causing more model damage, and ETD failing to capture high regularity corruptions. ViT overall showed better protection against medium and low regularity corruptions compared to ResNet. The aggregate metric of healed x utility is essential to penalize tasks in which either no unlearning occurred or model damage was significant. Failure to address either of the objective gets penalized via multiplication of the scores.
>
> ---
>
> Q3: While LLM unlearning is outside the scope of this work, the consistency Score (C-score) defined by Jiang et al. (2020) (Section 4, Eq. 1) is also applicable for LLMs, although expensive. There are several proxies that are cheap to compute (see C-score proxies in section 5 of Jiang et al.) such as “Pairwise Distance Based Proxies” and “Learning Speed Based Proxies”. For example, the unlearning paper of Zhao et al. successfully uses the learning speed proxy, which is a computationally cheap option, but requires tracking throughout the training run. Given this, our proposed framework is applicable to larger models too.
>
> ---
>
> W1: We are pleased to see that you see value in our taxonomy. Regarding the causal analysis ,we lay out the causes for method failures in section 3. To briefly summarize, the failure modes for different types of algorithms are as follows: SCRUB, BadT: reintroduction of corruptions via the retain set due to partial discovery since these methods use the retain set; Potion: low statistical regularity of corruptions leads to failure in methods that assume a concentration of corruptions in a few neurons; ETD: high statistical regularity of corruptions cannot be captured by methods that assume individual corruptions (e.g. noisy labels). We additional discuss the interplay of regularity and discovery rate, showing how for high regularity even at near perfect discovery rates corruptions get reintroduced via contaminated retain sets (and vice versa). We back these claims with experimental results for multiple unlearning methods that are representative for popular unlearning strategies (e.g., student-teacher via SCRUB). Our taxonomy, combined with the identified failure modes and their experimental backing provides a novel contribution to the literature, informing future method design. The design of REM was only possible due to the uncovered failure modes.
>
> ---
>
> W2: Thank you for pointing this out. We have added a “-” in step 2 to avoid ambiguity.
>
> ---
>
> W3: This is a good practical point. We expect that smaller representative datasets drawn from the full data will be a future direction of interest, which can be combined with any existing method. However, we point out that this issue is shared amongst all discussed methods, not unique to our method. Based on this, we hope the reviewer agrees that this is a direction that is orthogonal to, and outside the scope of our work
>
> ---
>
> W4: Thank you for pointing out this inconsistency. We have updated equation 2.
>
> ---
>
> Overall, we believe that we have addressed all of your questions and concerns. Reviewer UzFN, do you have any remaining concerns about the paper? We would love to discuss more and address them if so.

---

> > ### Comment · Reviewer_UzFN · 2025-11-22
> >
> > While several aspects of the rebuttal were not entirely convincing, the authors’ responses did address part of my concerns and reinforced the contribution of the work. I will therefore increase my score.
> > However, I do not think the revision merits an 8 at this stage, as some important limitations have yet to be sufficiently addressed.
> >
> > Q1. Fundamentally, the traditional definition of unlearning originates from the notion of the right to be forgotten, and therefore inherently enforces that the resulting model should resemble a retrained model. I still have doubts as to whether it is appropriate to regard this work purely as unlearning. Do the authors, then, believe that the traditional definition of unlearning needs to be expanded?
> >
> > W1. On page 1, author state: “This lack of a systematic understanding of when different methods fail or succeed is a fundamental blind spot as it hinders research progress and can lead to unpredictable failure in practice as shown in this work.”
> > Reading this paragraph, it gives the impression that the paper aims to discuss the causes of failure across unlearning methods in general, rather than the failure modes of existing unlearning methods specifically in the context of corrupted data. I wonder whether it might be appropriate to moderate the tone.
> >
> > W3. Regardless of the data composition assumed by other methods, I believe that accessing the full training dataset is never a realistic assumption in practical unlearning scenarios.

---

> > > ### Author Response · Authors · 2025-11-24
> > >
> > > Dear reviewer UzFN,
> > > Thank you for your quick response to our rebuttal. We are happy to hear that we could address some of your concerns, leading to a score increase.
> > >
> > > ---
> > >
> > > Q1: Our work also leads to a model that resembles a retrained model. The key question that needs expanding in the literature is “what kind of retrained model?” A model retrained excluding all undesired data or only the identified subset of undesired data? Our unlearned models resemble a retrained model trained on all data excluding the full set of corrupted samples. As we have shown, retraining without excluding all corrupted data reintroduces corruptions and is not a model that is satisfactory in practice. Unlearning therefore must be set in the context of a goal: “What do we want to unlearn?”. All identified corrupted data or all corrupted data. Based on the definition of what we want to unlearn, the goal of behaving like a model retrained without this data still holds. Prior works that study the same problem of unlearning include:
> > >
> > > Goel, Shashwat, et al. "Corrective Machine Unlearning." Transactions on Machine Learning Research (2024).
> > >
> > > Schoepf, Stefan, et al. "Potion: Towards Poison Unlearning." Journal of Data-Centric Machine Learning Research (2024).
> > >
> > >
> > > In summary, unlearning must be put into context by first defining what data we want to unlearn. Based on this information, we can then compare against a retrained model excluding this data. Depending on the metric of interest (e.g., MIA, accuracy), the retrained model can then be compared to the unlearned model to quantify the success of unlearning.
> > >
> > > ---
> > >
> > >
> > >
> > > W1: We are happy to rephrase this part to the following:
> > > “This lack of an understanding of when established unlearning methods fail or succeed is a fundamental blind spot as it hinders research progress and can lead to unpredictable failure in practice as shown in this work.”
> > > Does moving from “lack of systematic understanding of when different methods fail “ to  “lack of an understanding of when established unlearning methods fail” address your concern and better represent the contribution?
> > >
> > > ---
> > >
> > >
> > > W3: We agree with this practical point. Our motivation for using the full dataset is the following: Given that current methods are failing at the proposed tasks, even given the full training dataset, it is essential to first solve this easier setting before moving on to harder tasks with limited data availability. The full data results then also act as a benchmark for more restrictive settings. We lay the first step in this direction with the only method that performs strongly across our 2D space given the full training data.
> > >
> > >
> > > If we had a subset of the retain dataset rather than the entire retain dataset, it might even help with partial data unlearning (depending on what subset we have). Specifically, if the subset that we keep excludes the undiscovered data, we would actually be better off, as we forego the risk of reintroducing the corruptions when using the retain set. However, perhaps there are other subsets of the retain set that would hurt performance relative to using the whole retain set. This brings up an entire separate line of investigation on how to cleverly pick a (safe) subset of retain data to keep, which we hypothesize will have interesting interplay with the setting of partial discovery and regularity that we consider, as discussed above. But this is a completely separate endeavor that is orthogonal to our contribution (and can benefit several methods beyond just REM).
> > >
> > >
> > > We also wish to reiterate our contribution here, which is already substantial: We propose the first method that is able to address corrupted data unlearning tasks across different regularities and discovery rates while each prior method fails catastrophically on several configurations of corrupted unlearning tasks. This is already a significant step forward. We are looking forward to future work making the tasks even harder with more restrictive data availability. We are happy to add this explanation to the paper if you deem it helpful.

---

### Official Review · Reviewer_GrjL · 2025-10-27

**Soundness:** 3
**Presentation:** 2
**Contribution:** 2
**Rating:** 6
**Confidence:** 3

**Summary:**

This paper focuses on a particular machine unlearning setting, where the data is corrupted and the user only partially knows the corrupted data. In other words, some corrupted data still remain in the retain set. The authors propose a method to handle this setting, which can potentially handle cases with a high fraction of unknown corrupted data, as well as corruptions with high regularity. The method is based on Example-Tied Dropout (ETD), which introduces new parameters, and introduce random dropout/masking that is fixed for each example, such that the example-specific knowledge is encoded in these parameters (which can then be removed). The proposed method is largely similar to ETD, although there are certain differences. Results are reported on CIFAR10 with ResNet-9 and ViT with SVHN.

**Strengths:**

This paper has clearly identified a gap in the literature of machine unlearning, where the corrupted data discovery rate and regularity can pose a challenging scenario.

The proposed method shows good improvements under this corrupted data setting (Table 1).

**Weaknesses:**

The exact differences from ETD, and why these differences matter, is not very well explained. It seems to me that there are several differences, but I am not able to grasp well these implications, and why these differences are substantial or important.

The experiment results are compared against methods that are rather outdated. SCRUB and ETD seem to be the latest works in Table 1, but they are both works from 2023. I strongly suggest that the authors compare their works with more recent publications and methods. This would make their claims more convincing and substantial.

More specific questions are placed in the Questions section.

**Questions:**

The details of the NPO optimization method is unclear, and how it is applied to this classification is also not clear to me. Also, since the NPO optimization is to remove the impact of the forget data from the general parameters, can this be replaced by a more traditional unlearning method?

I am not familiar with the ETD method. How is it that so many parameters can be dropped out at test time, yet the performance is not severely affected? Is there some intuition for this?

It is unclear how the proposed method can handle cases of corrupted data with high regularity. Is it something to do with the masking strategy?

Could the authors clarify how they measure regularity exactly? Also, how do they vary the regularity in the data?

Did the authors experiment with different ways of adding parameters onto the model?
Also, how many parameters were added to the model (which are then removed)?


Overall, I currently recommend a borderline accept. However, I am not familiar with several key concepts in this paper as well as ETD, which is strongly related to the proposed method. I will revise my score based on the author’s responses.

---

> ### Author Response · Authors · 2025-11-19
> **1/3**
>
> Dear reviewer GrjL, thank you for your review and questions. We have tried our best to respond as thoroughly as possible and would be happy to have follow-up discussions.
>
> Q1: Great questions. Let us begin by clarifying that our method does not need to use NPO, necessarily. In step 2 of our approach (see pseudocode), we use an unlearning algorithm that relies on only the forget set, and NPO is a strong candidate for that, but we could have equally used gradient ascent, for example. As reported in Table 1, GA causes more model utility damage than NPO, therefore we chose to use the more stable NPO for REM (Note: NPO was specifically designed to avoid these collapses of GA).The idea here is to cleanse the core part of the network of any corruptions. Using unlearning algorithms that also rely on the retain set would not be suitable in this step, as the retain set contains undiscovered corruptions that would be reintroduced. Of course, avoiding the use of the retain set causes utility degradation, which is addressed in later steps. As stated in Section 5, "As research progresses, better UL methods that do not utilize $\mathcal{D}_{r}$ can replace NPO to boost REM performance further."
> Now, regarding NPO specifically, let us elaborate. NPO is an alignment-inspired objective that prevents the "catastrophic collapse" of model utility seen in Gradient Ascent by maintaining a reference to the original model parameters via a reference model. We adapt it for classification by making the simple change of replacing the NLP focused loss by the Cross-Entropy loss (Eq. 2). Hereby it provides the same benefit as in the LLM literature by preventing collapse compared to just using gradient ascent. To improve the reading experience, we will add the following into section 3 (where the other prior methods are presented): NPO (Zhang et al., 2024a) is an alignment-inspired loss function for unlearning that overcomes the catastrophic collapses of Gradient Ascent. This is achieved via a reference-model based loss calculation (equations taken from Zhang et al., 2024a):
>
> $L_{\text{GA}} = - \mathbb{E} \log(\pi_\theta(z))$
>
> $L_{\text{NPO}} = - \frac{2}{\beta} \mathbb{E} \log \sigma \left( - \beta \log \frac{\pi_\theta(z)}{\pi_{\text{ref}}(z)} \right)$
>
> Please let us know if this clarifies things.
>
> ---
>
>
> Q2: ETD splits the network parameters into a generalization partition, of which all neurons are always active, and a memorization partition, where each example has its own assigned “path” (dictated via an example-tied mask). Because the generalization neurons are always active during ETD training, this encourages any knowledge that is shared between examples to become encoded in the generalization partition, and only example-specific information to be encoded in the individual per-example memorisation paths. Because of this, dropping out the memorization partition doesn’t cause much damage to the model utility.
>
> ---
>
> Q3: Exactly, high regularity corruptions cannot be captured with ETD due to each sample having a unique random mask. This means that high regularity corruptions, where all corrupted data have a shared (generalizable) “concept”, like the poison trigger for example, would be learnt in the shared generalization neurons, thus our redirection would have failed. We overcome this with two intuitive tricks. First, we prevent learning of the corruptions in the generalization part of the model via our loss function that adds an NPO term on the generalization part while the redirection part of the loss allows for learning of the corruptions in the newly added neurons. Second, we assign all known corruptions the same mask, so that they can learn the shared “concept” (high regularity) in the same neurons in the newly added capacity, which would not be possible with a per-example mask strategy.
>
> ---
>
> Q4: We draw inspiration from the Consistency Score (C-score) defined by Jiang et al. (2020) (Section 4, Eq. 1). Conceptually, we define regularity in Equation 1. drawing inspiration from C-scores. In practice we have 3 levels of regularity which are monotonically increasing (by construction) but we have not measured empirically their regularity scores due to the thousands of models necessary for the calculation. We vary regularity by choosing tasks with inherently different levels of regularity: Random Label Swaps (Low due to random selection), Inter-class Confusion (Medium due to just two classes), and Poison Triggers (High due to exact same poison trigger for all corruptions). We leave it to future work to study finer-grained regularity levels rather than the 3 coarse regularity buckets that we study here. However, that's beyond the scope of our work, and our contribution is to show that this is an important axis that greatly affects behaviours of unlearning algorithms.

---

> > ### Author Response · Authors · 2025-11-19
> > **2/3**
> >
> > Q5: Great question, we believe there is exciting future work to be done here. In our work, we tried the simplest possible ideas which already worked very well. For ResNet we have used the same way to add parameters as the original ETD paper. For ViT we tested multiple options, finding that the FF layer in the transformer block is the most promising option. This aligns with findings on information storage in models (Geva, Mor, et al. "Transformer feed-forward layers are key-value memories." Proceedings of the 2021 Conference on Empirical Methods in Natural Language Processing. 2021.). We kept the addition of total parameters in line with the ETD paper for a fair comparison, using a 50/50 split. While this may sound like a lot of added parameters, this is no more than methods such as SCRUB which, like other student-teacher models, uses a full second model. In our case, we only ever have a subset of the expansion active that is selected by the masking strategy. The additional memory overhead is therefore simply a single mask size that is active during training.
> >
> > ---
> >
> > W1 While ETD formed the inspiration for our splitting the network into different parts, with different forward passes activating different parameters, ETD itself is substantially different, both in terms of its procedure as well as in terms of its goals and motivation. During the initial training (not post-hoc), ETD splits the network into a “generalization part”, which is always active, and a “memorization part”, of which each example has it’s own “mask”. The forward pass of a specific example thus activates all generalization neurons together with the example-specific memorization neurons. Samples that are memorized by a model (e.g., a noisy label that cannot be generalized) are more easily learned in the mask than in the generalization neurons, as the mask is less contested with information to be learned as it is not constantly active like the generalization part of the model. Indeed, we show that ETD can deal with low-regularity corruptions (i.e. memorized individual samples) quite well (an interesting discovery in its own right). But, as we discussed previously too, ETD is incapable of capturing higher-regularity corruptions.; an issue that we address in REM. REM’s key differences are the ability to handle high regularity corruptions and the ability to remove corruptions post-hoc without needing to interfere during the training process, which is limiting constraint of ETD. Achieving post-hoc unlearning is challenging, as corruptions are already encoded in the neurons and just finetuning with ETD would only reinforce the corruptions in their current locations, as this is the path of least resistance. Therefore, REM first removes the corruptions from the model using a retrain set free unlearning step (to avoid reintroduction of corruptions in partial discovery tasks) and then redirects the corruptions to the newly added capacity while simultaneously preventing reintroduction in the generalization part of the model via our custom loss that adds an NPO term on the generalization part of the model for corrupted data. This alone would not be enough to address high regularity corruptions. Therefore, we assign each known corruption the same mask, which allows for the collective learning of the regularity (e.g. a poision trigger) in shared neurons in the added mask capacity. Generally, REM is a multi-step procedure designed for corrupted data unlearning, which goes significantly above and beyond ETD and required the insights from our taxonomy to be designed to overcome prior method failures. We are happy to discuss further if there are other points we can clarify.

---

> > > ### Author Response · Authors · 2025-11-19
> > > **3/3**
> > >
> > > W2 We are comparing against the SOTA in corrupted data unlearning (Potion), which REM outperforms significantly across all tasks. We have now also added results for the method SalUn [ICLR24 (Spotlight)] (see results below). As we describe in the paper, prior methods fail in corrupted data unlearning due to their design that reintroduces corruptions into the model. These failures that we show with methods such as SCRUB (student-teacher distillation) are generally applicable.
> > >
> > >
> > > SalUn limits unlearning to selected weights and then applies an existing unlearning method, once again inheriting the drawbacks of existing methods. The original implementation uses fine-tuning on the retain set plus the randomly-relabeled forget set. This usage of the retain set will reintroduce corruptions as shown with our extensive experiments on SOTA methods. More importantly, while SalUn is SOTA at “forgetting” in privacy focused unlearning (where the goal is to avoid predicting the original label of the forget set), the random label finetuning aspect of SalUn makes it perform badly at “healing”, where for each example in the forget set, we want the unlearned model to now predict the correct label (healing) rather than just avoid predicting the originally-assigned incorrect label (forgetting). This algorithm is therefore not well suited for the problem we study. We report results for SalUn for the easiest setting of full discovery, showing that it already fails in this setting. Adding partial discovery would only further degrade the performance of SalUn. REM performance across all discovery levels is far better than SalUn on the easiest setting of 100% discovery. Results for ResNet with 1000 corruptions (analogous to the main paper without ETD pretraining):
> > >
> > >
> > > | UL type | Healed | Utility | Utility × Healed |
> > > | :--- | :---: | :---: | :---: |
> > > | REM | 81.16 ± 1.62 | 90.54 ± 0.15 | 73.40 ± 1.43 |
> > > | SalUn (100% discovery only) | 31.80 ± 7.88 | 91.50 ± 0.85 | 29.01 ± 6.98 |
> > >
> > > Generally, regarding the extensiveness of our empirical experimentation, we have used all unlearning methods from the original “Corrective Unlearning”(Goel et al)  benchmark and even extend upon it by adding additional methods, providing the to date most extensive benchmarking on corrupted data unlearning problems.
> > >
> > > ---
> > >
> > > We believe that this covers all families of models and state-of-the-art for the problem formulation that we study. Does the reviewer have a specific other method in mind that we should add?

---

### Official Review · Reviewer_PYtW · 2025-10-30

**Soundness:** 3
**Presentation:** 2
**Contribution:** 3
**Rating:** 6
**Confidence:** 3

**Summary:**

This paper introduces a method for unlearning of mislabelled data in the context of classification tasks. Authors first provide a problem definition and a analyse pros and cons of pre-existing works under a taxonomy covering two axes: levels of discovery of corrupted data and corruption patterns. Following this analysis, they propose a two-step unlearning method that borrows ideas from anti-memorisation technology to redirect corrupted information to a new set of dedicated weights. The approach shows more robustness across different scenarios than other methods.

**Strengths:**

The introduced taxonomy, analysis different forms of unlearning tasks in the context of classification problems is a nice framing of the problem. The analysis of a set of unlearning methods, including their strengths and reasons why they fail in certain scenarios was well written and helpful to a non expert in unlearning. Authors provide sensible and well justified arguments.

The proposed method heavily relies on prior work, but makes intelligent use of ideas developed in those work, which are designed for other problems, and adapts it to the problem at hand. The method is further designed in a way that builds on the analysis provided in the taxonomy section, making the paper more coherent as a whole.

**Weaknesses:**

-	One weakness refers to the inconsistent presentation quality of the paper, certain sections (such as section 4) are much better presented than others (such as section 5). The abstract and introduction in particular try to be too general, making it more difficult to understand the purpose and setting of the paper until section 2. Similarly, the paper mentioning a “universal unlearning method” yields expectations that the proposed work is not restricted to the classification problem.  Certain terms are discussed before being introduced, such as healing in the related works section.
Figures could also be improved, particularly figure 1 which requires accessing the appendix to be readable. In general, there are too many references to content in the appendix that is needed to read the main paper properly. Please refrain from including experiments in the main paper referring to results reported only in the supplementary materials. The main paper should be self-contained. Ablations are an important part of evaluating new methodology and results should be in the main paper.
-	Certain aspects of the methodology are confusing notably how the second fine-tuning stage prevents re-introducing corrupted information in the model. The claim that “undiscovered corrupted examples may be partially caught due to their masks partially overlapping” seems to be more wishful thinking than a rigorous scientific hypothesis.
-	In general, the methodological novelty is somewhat limited, being closer to an orchestration of pre-existing work.

**Questions:**

-	In general, I think this is an interesting contribution to the field of unlearning. The most interesting part of this work is the unlearning taxonomy and analysis of pre-existing work. I would recommend revising presentation in intro/abstract to facilitate reader understanding.
-	Why is NPO not discussed in section 4? It is a crucial part of the methodology in the following section. It would be great to have an explanation of how this approach works conceptually, as it is currently not discussed at all in the paper.
-	Could authors clarify what they mean when mentioning 50% and 100% capacity models? Is it just smaller models?

---

> ### Author Response · Authors · 2025-11-19
>
> Dear reviewer Pytw, thank you for your valuable comments.
>
> Q1: We appreciate your interest in our taxonomy and analysis of failure modes. We will happily put more emphasis on it for the camera-ready version, and place it more prominently in the abstract and the introduction. Are there any specific points you would like to see/recommend in the revision of the introduction/abstract?
>
> ---
>
> Q2: Thank you for raising this point. We only mentioned NPO in section 5 to stay within the page limit. Given the extra page for the camera ready version, we will add the following into section 3 (where the other prior methods are presented): NPO (Zhang et al., 2024a) is an alignment-inspired loss function for unlearning that overcomes the catastrophic collapses of Gradient Ascent. This is achieved via a reference-model based loss calculation ( we will add the original equations taken from Zhang et al., 2024a to section 3).
>
> As described in the paper, NPO can be replaced with any unlearning method that does not use the retain set (to avoid reintroduction of corruptions). We therefore expect REM to improve even further as better methods are developed that only utilize the forget set.
>
> ---
>
> Q3: Yes, 50% capacity is just a smaller version of the same model. We use this setup to allow the fairest possible comparison to ETD by using the best performing split (model capacity/added parameters) from their paper. As shown in our results, even in this case ETD completely fails at high regularity tasks (with the added “unfair” advantage of influencing the model during training vs REM which can be used post-hoc on any model).
>
> ---
>
> W1: Thank you for raising where we can improve readability. Given the 10th page allowed for the camera-ready version, we will move the ablations to the main paper and improve the writing in accordance with the outcomes of the rebuttal discussions from all reviews. Given the additional page, we will put the “A-F listed in appendix).” information from Fig. 1 in the caption, making it self contained. We are happy to alter the paper title to “... Towards a universal unlearning method for corrupted data in classification tasks” if this is technically possible.
>
> ---
>
>
> W2: Masking: The mechanism is based on gradient flow dynamics, not chance. By forcing all corrupted examples (Forget Set) to pass through the same active neurons in $\theta_{o2}$ (via the shared mask), we create a path of least resistance for minimizing the loss on these specific samples. In contrast, undiscovered corruptions in the Retain Set have random masks, but because they share the underlying high-regularity feature (e.g., the poison trigger) with the Forget Set, the gradients naturally steer this feature into the dedicated "channel" established by the Forget Set. The strong empirical results in Figure 1(g) across the regularity axis validate this mechanism. The ablations in the appendix further show that without the redirection of “step 3” performance suffers.
>
> To showcase these dynamics more explicitly, we have now additionally logged the accuracy of the model on the forget set with the additional neurons turned on/off during unlearning to show that the corruptions are contained within the additional neurons due to our masking strategy + custom loss. We have run an additional experiment to show the effectiveness of the routing (ResNet9, 50% discovery rate, poison trigger). The base model starts off with 99.0% accuracy on the corrupted data (i.e. nearly every sample adversely affected). During unlearning the base model (i.e. with no added parameters active) accuracy on corrupted data drops as follows over 5 epochs: [99.0 (start), 8.4, 11.6, 8.4, 7.6, 8.2]. 10% would be random chance with 10 classes. This is made possible because at the same time, the corruptions were rerouted into the newly added parameter partition, as shown by the accuracy on the corrupted data for the base model + the active parameters of the expansion: [11.4, 57.8, 99.0, 98.8, 99.8]. These results clearly show that the corrupted examples are indeed rerouted due to our masking + loss strategy.
>
> ---
>
> W3: We disagree that our work is “only” orchestration of pre-existing works. Due to the challenge of reintroduction of corrupted data via undiscovered corruptions, no trivial combination of prior works can address this problem. Our work develops a well thought out 4 step process with a non-trivial loss function that includes two different model versions, a redirection loss component, and a loss component to prevent reintroduction of corruptions. Putting this into perspective with NPO and its very recent successor SimNPO (NeurIPS 2025), NPO is DPO with one term left out of the equation and simNPO replaces one term in NPO with number of tokens in the prompt. Our methodology goes far beyond this and shows significant performance gains over prior works.
>
> ---
>
> Thanks again for the useful feedback. We are happy to discuss more if you have any follow up questions or comments.

---

### Official Review · Reviewer_cRxx · 2025-10-31

**Soundness:** 3
**Presentation:** 3
**Contribution:** 2
**Rating:** 4
**Confidence:** 4

**Summary:**

This paper studies machine unlearning for corrupted data in vision classifiers. Existing unlearning methods tend to be task-specific and fail when the corruption pattern changes. To diagnose these failures, the authors propose a 2-D taxonomy for unlearning tasks along two axes: Discovery Rate and Statistical Regularity. Through extensive experiments, the authors show that all prior unlearning algorithms succeed only in narrow regions of this 2D space, and fail catastrophically outside their specialized slices.To address this, the paper introduces REM (Redirection for Erasing Memory), redirecting all corrupted data into newly added neurons during unlearning, then drops these neurons so that all corruption effects are removed from the final model.

**Strengths:**

(1)The authors introduce 2D taxonomy to identify statistical regularity as a missing dimension in unlearning research.

(2)The authors proposed REM to redirect corrupted data into dedicated parameters and drop them, inspired by the existing method ETD.

(3)REM is the only method that avoids collapse across both dimensions, compared with baseline methods.

**Weaknesses:**

(1)EM doubles the model size and is not viable for real-world models. From the perspective of efficiency, the core design of REM expands the model by adding a full set of new parameters. This may be feasible for small CIFAR-scale networks, but it is not realistic for large scale models. Even the VIT structure meets the OOM problem. The paper does not discuss scalability, memory cost, or efficiency, which makes REM difficult to use in real-world unlearning scenarios.

(2)While the empirical results indicate that REM achieves strong performance across discovery levels and regularities, the paper does not provide any network-level evidence confirming that the proposed mechanism, i.e., redirecting corrupted information into θ₀₂ while keeping θ₀₁ clean, actually occurs inside the model.

(3)Incomplete reporting of ViT metrics. In Table 1, the authors report full results (Healed, Utility, and Utility×Healed) for ResNet-9, but only report the aggregated metric (Utility×Healed) for ViT. The omission of the individual Healed and Utility values for ViT makes the evaluation incomplete and prevents readers from assessing the unlearning–utility trade-off on a second architecture.

(4)Questionable OOM explanation for Potion–ViT experiments. The paper states that the Potion–ViT experiment resulted in Out-of-Memory on an A100 40GB GPU. This justification is not entirely convincing: Potion is not significantly more memory-intensive than several other baselines (e.g., SCRUB, BadT, NPO), all of which were successfully evaluated on ViT under the same setup. Moreover, techniques such as gradient checkpointing, reduced batch size, or accumulation are typically sufficient to avoid OOM on CIFAR-10–scale ViT models.

**Questions:**

(1)The main conceptual contribution of REM is the claim that corrupted examples can be explicitly routed into a newly added parameter partition. However, the paper provides no network-level analysis verifying that this mechanism actually occurs.

(2)Masking scheme lacks theoretical guarantees and may cause pathway overlap. The redirection mechanism relies on a mask assignment: all corrupted samples share one mask and each clean sample receives a random mask. However, the paper does not justify why this ensures correct sample separation.

---

> ### Author Response · Authors · 2025-11-19
> **1/2**
>
> Dear reviewer cRxx, thank you for your feedback.
>
> Q1 & W2: Thank you for raising this. To address this point, we have now logged the accuracy of the model on the forget set with the additional neurons turned on/off during unlearning to show that the corruptions are contained within the additional neurons. We have run an additional experiment to show the effectiveness of the routing (ResNet9, 50% discovery rate, poison trigger). The base model starts off with 99.0% accuracy on the corrupted data (i.e. nearly every sample adversely affected). During unlearning, the base model (i.e. with no added parameters active) accuracy on corrupted data drops as follows over 5 epochs: [99.0 (start), 8.4, 11.6, 8.4, 7.6, 8.2]. 10% would be random chance with 10 classes. This is made possible because at the same time, the corruptions were rerouted into the newly added parameter partition, as shown by the accuracy on the corrupted data for the base model + the active parameters of the expansion: [11.4, 57.8, 99.0, 98.8, 99.8]. These results clearly show that the corrupted examples are indeed rerouted, enabling unlearning without the problem of reintroduction that makes prior methods fail. We have added Fig. 11 in the appendix, as we cannot attach figures in the rebuttal text.
>
> Additionally, the paper already provides an ablation of all REM components in the appendix which shows how each component of the mechanism contributes to the overall performance, backing up the claims made for each step in REM. We will make sure to place this more prominently in the main paper with the additional 10th page allowed in case of acceptance.
>
> ---
>
> Q2: Design of masking scheme, guarantees, and pathway overlap. Thank you for bringing this up. Let us first clarify that pathway overlap not only is not a problem but is actually even helpful for unlearning.  Too much overlap in masks intuitively seems like a problem, as the model may learn generalized knowledge in the added capacity when a neuron is present in multiple masks of samples that share generalizable knowledge. Specifically, the issue would be that when the added capacity is dropped, this generalized knowledge will be lost and model utility drops. This can indeed be true if the masking strategy is active from the start of training (as is the case in standard ETD, for example; see lower utility of the ETD model vs the same size non-ETD model in Table 1). However, in the post-hoc unlearning setting of REM, the model is already trained. When we add additional capacity with a masking strategy and finetune the model with a mask per sample, existing generalization will not switch over to the added capacity, as the path of least resistance is to reuse the already present neurons in the model that contain this information. We therefore first need to remove the existing information from these neurons (the Remove step of REM) to be able to redirect them. If we were to then just finetune with a unique mask per sample, the generalizable corruption (in the case of high regularity corruption unlearning), e.g. the poison trigger, would once again be learned in the generalization part (unwanted!), as multiple samples can update these neurons vs in the added capacity where neurons receive few updates due to the masking strategy. Therefore, in order to redirect higher regularity corruptions (i.e. things that can be “generalized” such as a poison trigger) we require sufficient overlap of the masks on corrupted samples to “generalize” this unwanted information in the neurons of the added capacity instead of the generalization part of the model. We further improve the performance of this redirection via the constant NPO updates on the generalization part to prevent learning of corruptions in these neurons (see ablations in Table 3).  While we do not provide theoretical guarantees for the masking strategy, its design is carefully thought out to achieve unlearning of corrupted data of any regularity, and we show compelling empirical evidence of its success. Please let us know if any further questions on this topic remain.

---

> > ### Author Response · Authors · 2025-11-19
> > **2/2**
> >
> > W1: Thank you for raising this important point. First, let us clarify that we did not encounter any OOM errors with REM in our experiments. OOM happened with the Potion method during its parameter importance calculations (this is a known issue with a prior method, unrelated to REM). Next, we emphasize that while our current implementation of REM does double the total number of parameters, the number of active parameters is not doubled. This is because during training/inference, only one mask of the added parameters is active, not all added parameters. Further, student-teacher methods such as SCRUB actually double the active parameter count, as they need to forward pass through both a student and a teacher model. The same applies to NPO, which uses a reference model for the loss calculation. Therefore the NPO part of REM on the base model (without expansions) theoretically requires more active parameters than the computations on the expanded model (base+one mask < 2x base). Finally, we expect that with growing model size, the necessary relative size of the added neurons will diminish and depend on forget set size rather than model size itself. We view this exploration as being important future work that is outside of the scope of our paper. The key method contribution of this paper lies in “effectiveness”, creating the first method that can perform strongly across the entire 2D space of tasks, overcoming prior failure modes that no prior method could address. Our appendix already included section “A.8 COMPUTE TIME AND MEMORY”. We agree that future work into efficiency is of high interest now that we have addressed effectiveness. We look forward to future works exploring the scaling of REM style methods to larger models and additional modalities.
> >
> > ---
> >
> > W3: We report the full ViT results in the appendix (Table 5.), which are in line with our other findings. We will make sure to point out the additional detailed results more clearly in the main text.
> >
> > ---
> >
> > W4: Unlike NPO or REM which use first-order gradients, Potion requires computing parameter importance scores, which involves storing and computing over gradients for all parameters. On a ViT, which has a different parameter structure and scale than ResNet-9, this calculation exceeded the 40GB A100 memory in our implementation (using the Potion code from the official repo). We have discussed this with the authors of the Potion paper and it is a known issue. We believe that drastically improving the scalability of another method as beyond the scope of benchmarking and a possible future work by itself.
> >
> > ---
> >
> > Overall, we believe that our extensive clarifications and new results have thoroughly addressed your concerns. Reviewer cRxx, do you have any remaining concerns about the paper? We would love to discuss more and address them if so.

---

> > ### Comment · Reviewer_cRxx · 2025-11-26
> >
> > Thanks for your response. In your revised version, more clarification regarding the design of masking scheme, guarantees, and pathway overlap should be added.

---

> > > ### Author Response · Authors · 2025-11-27
> > >
> > > Dear reviewer cRxx, thanks a lot for your feedback and for confirming that the initial concerns you had raised are addressable by clarifying these points in our write-up, which we will do in the updated version by adding the provided rebuttal answers. We are wondering if you have any other concerns about the paper that prevent you from raising your score? Thanks a lot!

---

### Author Response · Authors · 2025-12-02
**Summary**

Dear AC, recognizing the heavy workload, here is a summary of the key points that were raised.

**Strengths listed by reviewers**

- **Novel Conceptual Framework:** We introduce a 2D taxonomy for unlearning tasks: Discovery Rate and Statistical Regularity. Reviewers agree this successfully explains why prior SOTA methods fail in specific "slices" of this space.
- **Universal Effectiveness:** There is consensus that REM is the first and only algorithm capable of performing strongly across the entire 2D task space, while each prior work fails catastrophically in all but one "slice" of the space of tasks we consider.
Strong Empirical Results: Our method achieves State-of-the-Art performance on standard benchmarks (CIFAR-10, SVHN, Imagenette) across different architectures (ResNet-9, ViT). “Backs claims with broad, rigorous benchmarks, showing SOTA scores and practical efficiency.”
- **Innovative Unlearning Mechanism**: Our core concept, redirecting corrupted data at unlearning time into newly added neurons and then discarding them, is viewed as a clever, effective adaptation of existing training-time "anti-memorization" techniques (like ETD) for the post-hoc unlearning setting.

**Concerns raised by reviewers**

- **Evidence for the "Redirection" Mechanism:** Reviewers asked whether our method actually isolates corrupted data in the new parameters or if this was just a theoretical claim without network-level evidence. **Update**: In addition to the existing ablations, we ran a new experiment logging model accuracy on the forget set with the added neurons toggled on/off. Results show accuracy on corruptions drops from 99% to ~8% (random chance with 10 classes) when our added parameters are deactivated, empirically proving the corruptions were successfully routed to and contained within the specific expandable parameters (new Fig. 5 in revised version).
- **Comparison to Outdated Baselines** One reviewer noted that baselines might be outdated and requested comparisons to more recent work. **Update:** We noted that we are comparing against the SOTA for this problem (Potion, 2024), and also added an additional comparison to SalUn (ICLR 2024 Spotlight). The results demonstrate that REM significantly outperforms SalUn, even in the easiest setting (100% discovery).
- **Model Efficiency & Memory:** Concerns were raised that REM doubles the model size (parameter inefficiency) and that OOM errors reported for the Potion baseline were unjustified. **Resolution:** We clarified that while total parameters increase, active parameters during training do not double (only one mask active at a time that increases active parameters), making computational cost lower than standard Student-Teacher methods (like SCRUB) that require twice the active parameters. We also clarified that Potion OOM errors are a known issue of this method and solving this is therefore not in the scope of our paper.
- **Differences from ETD:** Some reviewers found the distinction between REM and the prior work ETD (Example-Tied Dropout) unclear, questioning the novelty. **Resolution:** We clarified that ETD is different both in terms of  methodology and motivation. ETD is a training-time technique that fails on high-regularity corruptions (e.g., poison triggers) due to random masking. In contrast, REM is a post-hoc unlearning method that introduces a novel shared-masking strategy and specialized loss function. We demonstrated that this specific design, derived directly from our taxonomy's insights, is necessary to handle the full spectrum of regularity and partial discovery rates in a post-hoc setting, a feat no trivial combination of prior works can achieve.
- **Incomplete Metrics for Vision Transformers (ViT):** Reviewers incorrectly flagged that full "Healed" and "Utility" metrics were missing for ViT experiments. **Resolution:** We directed reviewers to the Appendix (Table 5 & Fig. 11 in the revised version) where full granular results for ViT are present. These results confirm the trends seen in ResNet.
- **Definition of Unlearning & Data Access:** One reviewer questioned if this counts as "unlearning" and argued that assuming access to the full training set is unrealistic. **Resolution:** We framed our unlearning problem as within the prior works of "Corrective Unlearning" and “Potion: Towards poison unlearning”, where the goal is a model that behaves as if trained without corruptions. We argued that solving the “easier” problem with full data access is a necessary prerequisite before tackling partial-data scenarios, as no prior method succeeded even with full data.

We appreciate the feedback of the reviewers and believe that our paper provides two strong contributions via our **(1) contribution to knowledge via the 2D taxonomy**, as well as our **(2) methodological contribution, REM, as the first method overcoming the identified failure modes**. We believe that the responses summarized above fully address all concerns raised by the reviewers.

---

### Meta-Review · Area_Chair_TVDS · 2026-01-07

**Summary:**

Reviewers consistently recognized the value of the paper’s two main contributions: (i) a clear and useful 2D taxonomy that systematizes corrupted data unlearning tasks and explains why existing methods tend to work only in narrow regimes, and (ii) a redirection-based unlearning method that demonstrates robust empirical performance across a range of evaluated settings. The empirical results were generally viewed as solid.

At the same time, reviewers raised questions about the clarity and justification of the proposed mechanism, its distinction from closely related work such as ETD, the strength of the underlying assumptions (e.g., full data access and excluding retraining), and the scope and tone of some claims, which were perceived as overly strong. Additional concerns regarding presentation clarity, baseline completeness, and exposition were also pointed out. Overall, these concerns primarily relate to wording and positioning rather than the technical validity of the core contributions.

**Reviewer Concerns:**

The rebuttal addressed several of the main technical concerns. Specifically, the additional experiments and clarifications provided stronger evidence that corrupted information is redirected to newly added parameters and removed after deactivation, reducing concerns that the mechanism is purely heuristic. The authors also clarified the differences between REM and prior methods such as ETD, highlighting the post-hoc unlearning setting, the masking design, and different failure behaviors, which helps address questions about novelty. Finally, comparisons with more recent baselines were included, addressing concerns about outdated evaluations.

Some concerns remain partially outstanding. Reviewers pointed out that assumptions such as full access to the training data may limit practical realism, as well as that claims regarding universality or practical safety may be too strong given the stated scope. The paper would benefit from clearer positioning of claims to avoid misleading.

**Reviewer Scores:**

Reviewer cRxx: probably an increase in score, as the rebuttal provided concrete additional evidence and clarifications.

Reviewer PYtW: Probably unchanged or a little bit increase. The reviewer’s primary concerns about mechanism clarity and overgeneralization were largely addressed. The incremental nature of the method may still limit a larger score increase.

Reviewer GrjL: Probably an increase, given that many questions stemmed from unclear presentation/related works, and the rebuttal clarified these.

Reviewer UzFN: Reviewer joined the discussion and already indicated a score increase after the rebuttal. Reviewers also pointed out there are concerns remained – likely a small increase rather than a big one.

---

### Decision · Program_Chairs · 2026-01-26

Accept (Poster)